# Microglial NF-κB drives tau spreading and toxicity in a mouse model of tauopathy

Chao Wang [1], Li Fan[2], Rabia R. Khawaja[3], Bangyan Liu[2], Lihong Zhan[1], Lay Kodama [2], Marcus Chin [1], Yaqiao Li[1], David Le[1], Yungui Zhou[1], Carlo Condello [4,5], Lea T. Grinberg [5,6], William W. Seeley[5,6], Bruce L. Miller[5,6], Sue-Ann Mok [7], Jason E. Gestwicki [4,8], Ana Maria Cuervo [3], Wenjie Luo [2 ✉] & Li Gan [1,2 ✉]

Activation of microglia is a prominent pathological feature in tauopathies, including Alzheimer's disease. How microglia activation contributes to tau toxicity remains largely unknown. Here we show that nuclear factor kappa-light-chain-enhancer of activated B cells (NF-κB) signaling, activated by tau, drives microglial-mediated tau propagation and toxicity. Constitutive activation of microglial NF-κB exacerbated, while inactivation diminished, tau seeding and spreading in young PS19 mice. Inhibition of NF-κB activation enhanced the retention while reduced the release of internalized pathogenic tau fibrils from primary microglia and rescued microglial autophagy deficits. Inhibition of microglial NF-κB in aged PS19 mice rescued tau-mediated learning and memory deficits, restored overall transcriptomic changes while increasing neuronal tau inclusions. Single cell RNA-seq revealed that tau-associated disease states in microglia were diminished by NF-κB inactivation and further transformed by constitutive NF-κB activation. Our study establishes a role for microglial NF-κB signaling in mediating tau spreading and toxicity in tauopathy.

[1] Gladstone Institutes, University of California, San Francisco, San Francisco, CA, USA. [2] Helen and Robert Appel Alzheimer's Disease Institute, Brain and Mind Research Institute, Weill Cornell Medicine, New York, NY, USA. [3] Department of Developmental and Molecular Biology, Albert Einstein College of Medicine, Bronx, NY, USA. [4] Institute for Neurodegenerative Diseases, University of California, San Francisco, San Francisco, CA, USA. [5] Department of Neurology, University of California, San Francisco, San Francisco, CA, USA. [6] Memory and Aging Center, University of California, San Francisco, San Francisco, CA, USA. [7] Department of Biochemistry, Faculty of Medicine and Dentistry, University of Alberta, Edmonton, AB, Canada. [8] Department of Pharmaceutical Chemistry, University of California, San Francisco, San Francisco, CA, USA. ✉email: wel2009@med.cornell.edu; lig2033@med.cornell.edu

Abnormal aggregation and spreading of the microtubule-associated protein tau is the key defining feature of a group of heterogeneous neurodegenerative diseases known as tauopathies[1]. Alzheimer's disease (AD) is the most common tauopathy, with a hallmark of neurofibrillary tangles (NFTs) composed of insoluble tau fibrils. Tau pathology correlates more closely with synaptic loss, neurodegeneration, and cognitive decline than does amyloid pathology[2–4]. Understanding the pathogenic mechanism induced by pathological tau is critical for developing effective therapeutics for AD and other tauopathies.

Neuroinflammation, the immune response in the central nervous system (CNS) characterized by reactive gliosis and increased inflammatory molecules, is one of the early and sustained pathological features of tauopathies[5]. Microglia, the resident innate immune cells in the CNS, are key players in neuroinflammation. Recent genetic and gene network analysis of late-onset AD (LOAD) identified several risk variants predominantly expressed in microglia, implicating a pivotal role of microglia in AD pathogenesis[6]. Reactive microglia have been observed to associate with NFTs in AD[7], primary tauopathies[8,9], and tau transgenic models[10,11]. In vitro studies confirm that tau directly activates microglia and triggers a proinflammatory profile[12]. Accumulated evidence also suggests that microglia are involved in tau-mediated pathobiology, including tau phagocytosis[13,14], post-translational modification and aggregation[15,16], spreading[17], and tau-induced synaptic loss[18]. Blockade of microglia proliferation attenuates tau-induced neurodegeneration and functional deficits[19]. However, the molecular pathways underlying microglia-mediated tau toxicity remain poorly defined.

NF-κB is a transcription factor known to modulate many target genes that are associated with neuroinflammation, glial activation, oxidative stress, cell proliferation, and apoptosis in the central nervous system. IκB kinase (IKK) activates NF-κB via phosphorylation and subsequent degradation of IκBα, an inhibitor of NF-κB[20]. Genetic manipulations of IκB and IKK using the Cre-lox system enable conditional activation or inactivation of NF-κB and the dissection of functions of NF-κB in specific cell types[21]. Such studies have shown that neuronal NF-κB plays an essential role in synaptic plasticity and regulating learning and memory behaviors in basal conditions[22,23], whereas in pathological conditions, the anti-apoptotic role of neuronal NF-κB is associated with neuroprotective effects[24,25]. Microglial NF-κB also regulates synaptic plasticity[26].

Dysregulation of NF-κB has been implicated in AD pathogenesis[27,28]. Indeed, in a meta-analysis, NF-κB signaling was found to be among the most perturbed pathways in LOAD brains[29]. NF-κB is known to be activated by amyloid β (Aβ) and to contribute to Aβ production[30,31]. We previously showed that specific inhibition of microglial NF-κB activation via deacetylation of p65 protected against Aβ toxicity in glial-neuron co-cultures[32]. However, very little is known about the role of microglial NF-κB activation in tauopathy.

Our current study establishes NF-κB signaling as a central transcription factor driving tau responses in vitro and in vivo models of tauopathy. By genetically deleting or activating IKKβ kinase selectively in microglia, we investigated how microglial NF-κB signaling contributes to tau processing, seeding, and spreading, as well as tau toxicity using behavioral studies. Using bulk and single-nuclei RNA-sequencing (RNA-seq), we dissected how microglial NF-κB activation and inactivation modify overall transcriptomic changes, tau-associated microglial states, and underlying pathways in microglia. Our findings uncover mechanisms by which microglial NF-κB activation drives disease progression in tauopathy.

## Results

**Tau activates NF-κB pathway in microglia.** To determine tau-induced transcriptome changes in microglia, we treated primary mouse microglia with synthetic 0N4R full-length wild-type (FL_WT) tau fibrils for 24 h and analyzed them by RNA-seq. Synthetic recombinant tau monomers and fibrils contained negligible endotoxin (Supplementary Data 1). Out of 2975 differentially expressed genes (DEGs, False Discovery Rate (FDR) < 0.05) (Fig. 1a and Supplementary Data 2), Ingenuity Pathway Analysis (IPA) revealed that the top affected canonical pathways were associated with cellular immune responses, morphological changes, cell movement, survival, and proliferation (Fig. 1b, Supplementary Data 3). NF-κB signaling, which is involved in the upstream or downstream regulation of many other canonical pathways, such as Tumor Necrosis Factor Receptor 2 (TNFR2), toll-like receptor (TLR), interferon, and interleukin 12 (IL-12) signaling, was among the top altered cellular immune response pathways. Many NF-κB target genes were upregulated, including chemokines and receptors, such as C–C Motif Chemokine Ligand 5 (Ccl5), C–X–C Motif Chemokine Ligand 9 (Cxcl9), complement component C3, proinflammatory cytokines interleukin 1 beta (Il1b), Interleukin 12b (Il12b), and Tumor Necrosis Factor (Tnf), Fc fragment of IgG receptors involved in phagocytosis such as Fc receptor, IgG, high affinity I (Fcgr1), and the NF-κB pathway component NF-κB Inhibitor Alpha (Nfkbia) (Fig. 1a).

NF-κB transactivation was also measured with a reporter assay. Primary microglia were infected with lentivirus expressing EGFP under the control of the 5× κB enhancer element (Lenti-κB-dEGFP)[32]. Both FL_WT and K18/PL tau fibrils, a truncated form of human tau fibrils containing only microtubule-binding domains with the P301L MAPT mutation[33], significantly induced EGFP expression, indicating the activation of NF-κB promoter by tau fibrils (Fig. 1c, d). Wild-type or P301L mutant tau monomers also induced EGFP expression in microglia infected with Lenti-κB-dEGFP, confirming that both tau fibrils and monomers can activate the microglial NF-κB pathway (Supplementary Fig. 1a, b). The extent of NF-κB activation induced by 2–2.5 ug/ml tau fibrils was comparable to that of 50 ng/ml lipopolysaccharide, which contains 250–2000 fold more endotoxin (Supplementary Data 1).

We next profiled microglial transcriptional changes in a tauopathy mouse model of isolated microglia from 11-month-old PS19 mice. Disease-associated microglia (DAM) are identified as a subset of microglia associated with AD and other neurodegenerative diseases with a unique transcriptional signature[34,35]. Some DAM signature genes, such as Cst7, Axl, Lpl, Itgax, Clec7a, Cox6a2, Ank, Csf1, and NF-κB target genes, such as Tnf, Il1rn, Tlr2, and Traf3, were among the upregulated DEGs (FDR < 0.05, Supplementary Data 4) (Fig. 1e). Fifty-nine of 187 upregulated DEGs in PS19 microglia, including DAM genes (e.g., Clec7a, Cst7, Lpl) and NF-κB target genes (e.g. Tnf, Il1rn, Tlr2), were also upregulated in tau fibrils stimulated microglia, indicating a similarity between tau-induced transcriptomics changes in microglia in vitro and in vivo. (Fig. 1a, e and Supplementary Data 5). NF-κB signaling was also identified as one of the top differentially regulated pathways in PS19 microglia (Fig. 1f). Other pathways and DEGs included cell growth and death, mitochondrial dysfunction, and autophagy pathways, together with cytochrome-c oxidases, the terminal enzymes of the mitochondrial respiratory chain, such as Cox6a2, Cox8a, and endo-lysosome associated genes, such as Ctsb, Lamp1, Grn, Rab34, all of which were upregulated in PS19 microglia (Fig. 1e, f and Supplementary Data 6). Consistent with the canonical pathways associated with NF-κB activation, cell movement, and migration, inflammatory responses and phagocytosis were the top activated biological functions in tau-stimulated microglia and

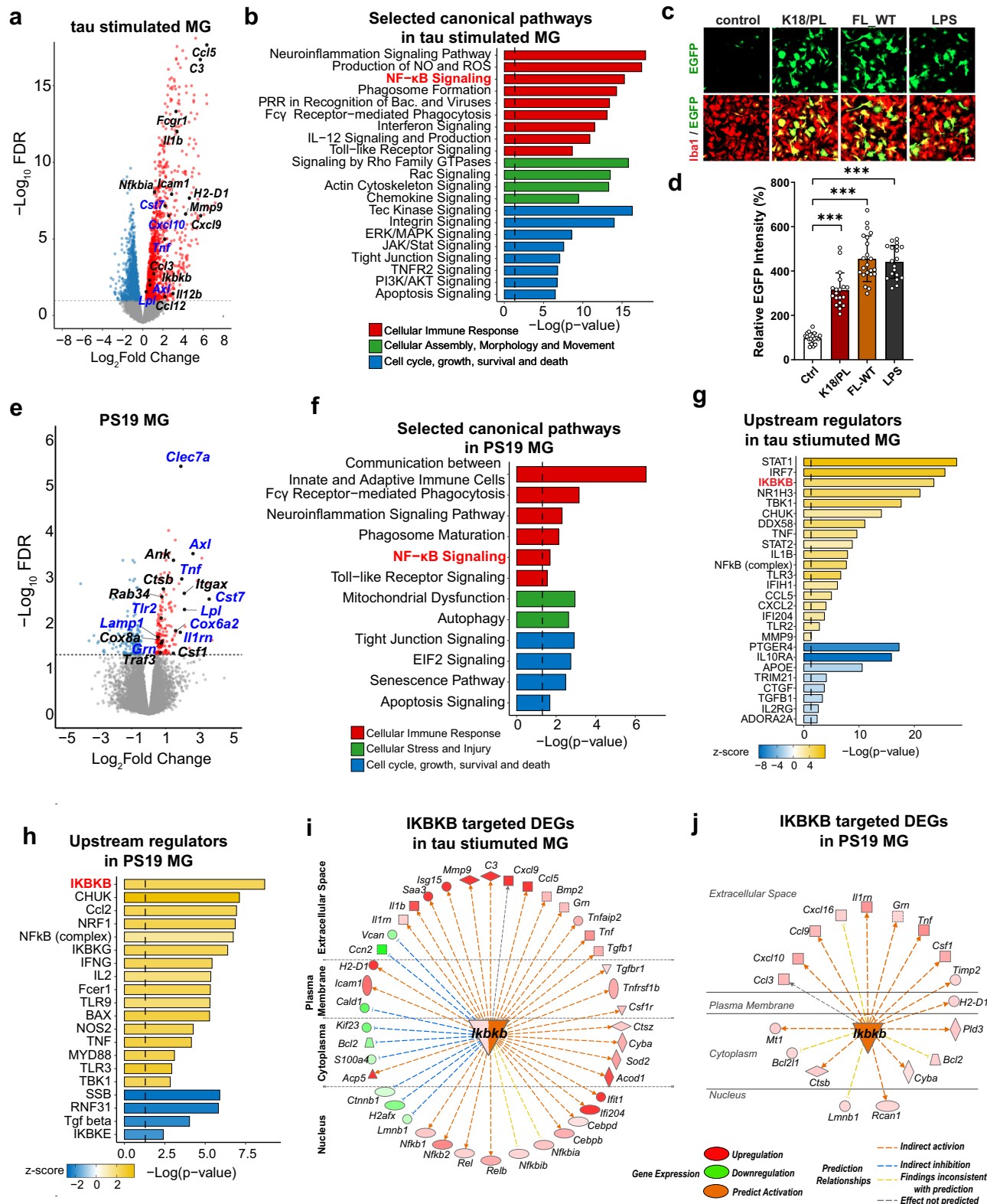

**a** tau stimulated MG

**b** Selected canonical pathways in tau stimulated MG

**c** control   K18/PL   FL_WT   LPS

**d**

**e** PS19 MG

**f** Selected canonical pathways in PS19 MG

**g** Upstream regulators in tau stiumuted MG

**h** Upstream regulators in PS19 MG

**i** IKBKB targeted DEGs in tau stiumuted MG

**j** IKBKB targeted DEGs in PS19 MG

PS19 microglia (Supplementary Fig. 1c, d). Specifically, IκB kinase complex subunit IKKβ (*Ikbkb*) was identified as one of the top upstream regulators responsible for tau-mediated transcriptomic changes in primary microglia (Fig. 1g) and PS19 microglia (Fig. 1h) using IPA upstream regulatory analysis[36]. *Ikbkb* gene itself was also upregulated in tau-stimulated microglia, and predicted to be activated in PS19 microglia (Fig. 1i, j). Indeed, a large repertoire of DEGs was predicted to be regulated by IKKβ, supporting IKKβ activation as a master regulator of tau-mediated microglial NF-κB activation.

**NF-κB transforms transcriptomes in cultured microglia.** To directly investigate the cell-autonomous effects of NF-κB in

**Fig. 1 Tau activates NF-κB pathway in microglia. a** Volcano plot of significant DEGs between full-length wild-type tau fibril-treated and vehicle-treated primary mouse microglia. Colored points represent FDR < 0.05 (−log (FDR) ≥ 1.3, dashed line), log₂FC > 0 upregulated genes (red) and log₂FC < 0 downregulated genes (blue). Selected NF-κB pathway associated genes and DAM signature genes are highlighted. FC fold change. **b** Selected IPA canonical pathways identified for significant DEGs of tau-stimulated microglia. Canonical pathways are grouped by indicated categories. **c, d** Primary microglia infected with NF-κB reporter (Lenti-κB-dEGFP) virus were incubated with K18/PL fibrils (2.5 ug/ml), full-length wild-type (FL-WT) tau fibrils (2 ug/ml), and LPS (50 ng/ml) for 24 h. **c** Representative fluorescence high-content images of EGFP (green) and microglial marker Iba1 (red); **d** Quantification of EGFP intensity. Scale bar, 50 μm. Values are mean ± SD, relative to vehicle control. Total N = 18(ctrl), 19(K18/PL), 22(FL-WT) and 19(LPS) wells from three independent experiments. P-value was calculated using multilevel mixed-effect model with experiment as hierarchical level, ***p < 0.001. Source data are provided as a Source data file. **e** Volcano plot of significant DEGs between isolated microglia from 11-month-old PS19 mice and non-transgenic controls. Selected NF-κB pathway associated genes and DAM signature genes are highlighted. **f** Selected IPA canonical pathways identified for significant DEGs in PS19 microglia. Canonical pathways are grouped by indicated categories. **g, h** Selected IPA predicted upstream regulators for DEGs of tau-stimulated microglia (**g**) and PS19 microglia (**h**). IKBKB (distinguished in red) is among the top upstream regulators. **i, j** The network of DEGs regulated by IKBKB in tau-stimulated microglia (**i**) and PS19 microglia (**j**). Locations, gene expression levels and predicted relationships with IKBKB are illustrated as indicated labels. Shades of red and green represent log₂FC of selected DEGs. Genes highlighted in blue in **a** and **e** are selected DEGs upregulated in both tau-stimulated microglia and PS19 microglia. P-values of **b**, **f**, **g**, **h** were calculated using right-tailed Fisher's exact test with threshold of significant enrichment as p-value ≤ 0.05 (indicated by a dotted line of −log (p-value) = 1.3).

microglia, we activated IKKβ in microglia by crossing *Cx3cr1^CreERT2* mice[37] with R26-Stop^FLikk2ca mice, in which a constitutively active form of IKKβ was inserted into the Floxed-Rosa locus with a stop codon (hereafter referred to as "*IkbkbCA^F/F*" mice)[38]. We treated primary microglia from *Cx3cr1^CreERT2/+*; *IkbkbCA^F/F* mice with 4-hydroxy tamoxifen[39] to induce Cre expression (Fig. 2a). Elevation of *Ikbkb* was confirmed by RT-qPCR (Fig. 2b). We compared the transcriptomes induced by IKKβCA with those induced by tau stimulation, and observed 693 shared DEGs (556 upregulated and 137 downregulated) (Fig. 2c and Supplementary Data 7), suggesting that tau-induced alterations may be partially mediated by NF-κB activation. IPA analyses of shared DEGs showed that in tau-stimulated microglia, enhanced cell proliferation, movement, phagocytosis, cytotoxicity, and reduced cell death (Fig. 2d), as well as elevation of immune response pathways such as interferon, TLR signaling, and inhibition of apoptosis signaling (Fig. 2e and Supplementary Data 8), may be mediated through NF-κB activation.

In complementary experiments, we selectively inactivated NF-κB in microglia by crossing *Cx3cr1^CreERT2* mice with *Ikbkb^F/F* mice[40], and treated primary microglia with 4-hydroxy tamoxifen[39] to induce Cre expression (Fig. 2f). Deletion of *Ikbkb* in microglia was confirmed by RT-qPCR (Fig. 2g). To further examine NF-κB-dependent microglial transcriptomes, we next compared the DEGs between IKKβ null and IKKβCA microglia. Inactivation of NF-κB in microglia resulted in 208 unique DEGs (78 upregulated and 130 downregulated), while activation resulted in 1381 unique DEGs (775 upregulated and 606 downregulated) (Fig. 2h, Supplementary Data 9). IPA analysis revealed that inactivation of microglial NF-κB led to pathways associated with elevated cell death, and decreased cell movement and proliferation. In direct contrast, activation of microglial NF-κB altered pathways associated with decreased cell death, but elevated cell movement, proliferation, and phagocytosis (Fig. 2i, j). Analyses of the top affected canonical pathways revealed that those unique for IKKβ null microglia were associated with cell cycle regulation, nucleotide biosynthesis, and DNA damage repair, whereas those unique for IKKβCA were enriched for integrin, TNFR2, and PI3K/Akt signaling (Supplementary Fig. 2a, b and Supplementary Data 10). Surprisingly, ~400 DEGs were shared by activating and inactivating microglial NF-κB (Fig. 2h and Supplementary Data 11). Among the shared DEGs, a great fraction was involved in interferon signaling (Supplementary Fig. 2c and Supplementary Data 10). Specifically, interferon regulatory factor 3 and 7 (*Irf3*, *Irf7*), interferon-α/β receptor (*Ifnar*), and interferon-γ (*Ifng*) were the top upstream regulators predicted to be activated for these transcriptomic changes (Supplementary Fig. 2d). These results support the extensive crosstalk between interferons and the NF-κB pathway in microglia[41].

**NF-κB promotes microglial processing and release of tau species with seeding activity.** In cultured conditions, we found that tau fibrils can be readily taken up by microglia, but not neurons, in a time-dependent manner, and once inside the cell, they colocalize with the late endosome/lysosome labeled by Dextran[42] (Supplementary Fig. 3a, b), where they can be proteolytically processed[43,44]. To determine the effects of NF-κB signaling on this process, we examined tau fibril accumulation in IKKβ null and IKKβCA microglia by comparing them to their respective littermate wild-type controls (*Ikbkb^+/+* and *Ikbkb^WT*). Inactivation of NF-κB enhanced, while activation diminished, the number of tau fibrils remaining in microglia (Fig. 3a, b). To dissociate uptake from clearance, we performed a pulse-chase assay by preloading microglia with tau fibrils and assessing the time-dependent clearance in the next 24 h (Fig. 3c). Compared with corresponding littermate wild-type controls, inhibition of NF-κB slowed down, while activation accelerated, tau clearance in microglia (Fig. 3d–g).

Next, we isolated sarkosyl-insoluble tau (AD-tau) from postmortem human AD frontal cortical tissue[44] (Fig. 3h). After incubation with microglia, AD-tau was internalized and detected by phosphorylated tau (pTau) antibody AT8 as intracellular puncta (Fig. 3i, j). After chasing in tau-free medium for 24 h, ~23% (±7.3%) of total AT8-positive-pTau taken up by microglia was released in the conditional medium (CM) and ~22.5% (±5.2%) remained in the cells (Fig. 3k), suggesting that ~55% internalized pTau were degraded or processed by microglia.

We next examined the seeding activity of microglia-released tau using HEK human tau RDP301S biosensor cells[45]. The AD-tau microglial conditional medium (AD-tau CM) induced intracellular tau aggregation in HEK biosensor cells while CM of microglia without tau loading did not (Control CM) (Fig. 3l, m), demonstrating a potent seeding activity of AD-tau processed and secreted by microglia. Moreover, NF-κB inhibitor TPCA-1[46] significantly reduced AT8-positive pTau released from microglia (Fig. 3n) with a modest increase of intracellular pTau (Fig. 3o). Thus, inhibition of NF-κB reduces the microglial processing and release of pathological tau species.

**Microglial NF-κB activation promotes tau seeding and spreading in PS19 mice.** Tau pathology spreads from the entorhinal cortex to the hippocampal region in the early stage of AD[47]. To model the seeding and spread of tau inclusions in vivo, we inoculated young PS19 mice with exogenous tau seeds as described previously[33] (Supplementary Fig. 4a). To investigate the role of microglia, we depleted microglia by feeding young PS19 mice a diet containing the colony-stimulating factor 1 receptor

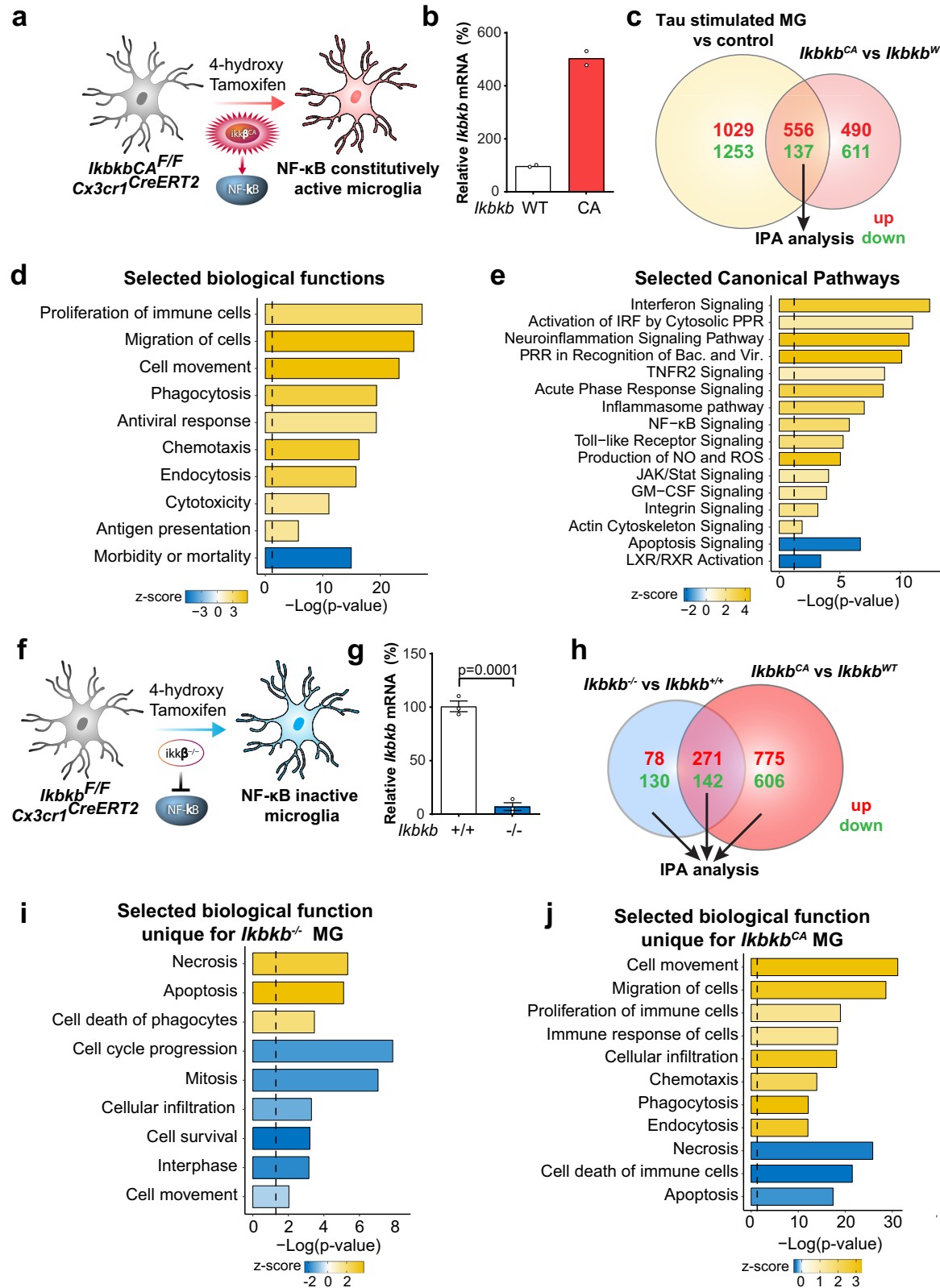

inhibitor PLX5622 (PLX)[48], followed by inoculating with either brain extract from progressive supranuclear palsy (PSP) patients (Supplementary Fig. 4b) or synthetic K18/PL tau fibrils (Fig. 4a) unilaterally into the hippocampus. K18/PL tau fibrils-injected mice were continuously fed with PLX for one month to prevent microglial repopulation, which reduced the number of microglia by over 80% (Supplementary Fig. 4c, d). Depletion of microglia

significantly reduced the number of tau inclusions in AT8+ neurons in both contralateral and ipsilateral cortex of PS19 mice inoculated with PSP brain extract, consistent with the notion that microglia promote tau seeding and spreading in mouse tauopathy models[17] (Supplementary Fig. 4e, f). Similarly, microglial depletion significantly reduced the seeding and spreading of tau in PS19 mice inoculated with synthetic K18/PL tau fibrils, which

**Fig. 2 NF-κB-dependent transcriptomic reprograming in primary microglia. a** Experimental diagram illustrating *IkbkbCA^{F/F} Cx3cr1^{CreERT2}* primary microglia incubated with 4-hydroxytamoxifen to induce IKKβCA expression. **b** Increased *Ikbkb* mRNA levels were confirmed by quantitative PCR analysis. *N* = 2 per genotype. **c** Venn diagram comparing DEGs between *Ikbkb^{CA}* microglia and tau-stimulated microglia. DEGs: FDR < 0.05 in comparison to corresponding controls. **d, e** Selected IPA biological functions (**d**) and canonical pathways (**e**) identified for shared DEGs of *Ikbkb^{CA}* microglia and tau-stimulated microglia identified in **c. f** Experimental diagram illustrating *Ikbkb^{F/F} Cx3cr1^{CreERT2}* primary microglia incubated with 4-hydroxytamoxifen to delete IKKβ. **g** Quantitative PCR analysis of *Ikbkb* deletion. *N* = 3 per genotype. Values are mean ± SEM, relative to vehicle control, two-tailed *t*-test. **h** Venn diagram comparing DEGs between *Ikbkb^{−/−}* and *Ikbkb^{CA}* microglia. DEGs: FDR < 0.05 in comparison to corresponding wild-type control, respectively. **i, j** Selected IPA biological functions identified for unique DEGs of *Ikbkb^{−/−}* microglia (**i**) and *Ikbkb^{CA}* microglia (**j**). *P*-values of **d, e, i, g** were calculated using right-tailed Fisher's exact test with threshold of significant enrichment as *p*-value ≤ 0.05 (indicated by a dotted line of −log (*p*-value) = 1.3). Source data are provided as a Source data file.

induced tau spreading within 1 month (Fig. 4a–c). Importantly, no AT8+ or MC1+ neurons were detected in PS19 mice inoculated with PBS, or in non-transgenic control mice inoculated with tau seeds (Supplementary Fig. 4g).

Since microglial processing of tau is regulated by NF-κB, we reasoned that microglial NF-κB activity could also affect tau seeding and spreading in vivo. We selectively deleted *Ikbkb* in adult microglia of PS19 mice by crossing *Cx3cr1^{CreERT2}* with *Ikbkb^{F/F}* and PS19 mice. Tamoxifen injection diminished IKKβ expression in adult microglia from *Cx3cr1^{CreERT2/+}; Ikbkb^{F/F}* (referred to as *Ikbkb^{−/−}*) mice, as confirmed with qRT-PCR analyses (Fig. 4d, e). To measure how tau seeding and spreading were affected, 3-month-old mice were inoculated with tau fibrils in one side of the hippocampus 2 weeks following tamoxifen injection (Fig. 4f). K18/PL fibrils were used to induce tau spreading since only 4 weeks post-inoculation time is needed for robust wide-spread tau propagation. Inactivation of microglial NF-κB reduced the amount of MC1+ tau inclusions significantly at the ipsilateral side of the cortex (Fig. 4g, h), similar to the effect of depleting microglia (Fig. 4c), suggesting a critical role for microglial NF-κB activation in tau spreading. In complementary experiments, tamoxifen injection enhanced IKKβ expression in adult microglia from *Cx3cr1^{CreERT2/+}; IkbkbCA^{F/F}* (referred to as *Ikbkb^{CA}*) mice (Fig. 4i, j). RNA-seq analysis of upregulated DEGs in *Ikbkb^{CA}* mice brain (Supplementary Data 12) confirmed that NF-κB is the top transcription factor that leads the transcriptomic changes (Supplementary Fig. 5). In complementary experiments, we tested if activating microglial NF-κB could enhance tau propagation. To avoid ceiling effects, we reduced the amount of inoculated tau fibrils to that in microglial-depletion experiments. One month after the inoculation, *Ikbkb^{CA}* mice exhibited elevated tau inclusions significantly on the ipsilateral side, with a modest increase on the contralateral side of the hippocampus (Fig. 4k–m). Our findings indicate that NF-κB activation is essential in promoting microglial-mediated tau seeding in vivo.

**Inactivation of microglial NF-κB partially restores microglia homeostasis and protects against spatial learning and memory deficits in PS19 mice.** Microgliosis and amoeboid morphological changes are early and sustained phenomena in PS19 mice[10]. Inactivation of microglial NF-κB reduced microgliosis in the hippocampus of 8–9-month-old PS19 mice (Fig. 5a, b), and a similar trend was observed in the cortex (Fig. 5c, d). Imaris analysis of microglial morphology further revealed that inhibition of microglial NF-κB resulted in more ramified morphology with longer processes and more branches, partially reversing the amoeboid morphology induced by pathogenic tau (Fig. 5e–g).

PS19 mice exhibit spatial learning and memory deficits starting at 7–8 months of age[49,50]. To examine the functional outcome of inhibition of microglial NF-κB activity in PS19 mice, we tested 8–9-month-old *Ikbkb^{+/+}, Ikbkb^{−/−}, Ikbkb^{+/+}; P301S+*, and *Ikbkb^{−/−}; P301S+* mice in the Morris water maze (MWM) test, a hippocampus-dependent assay that evaluates spatial learning and

memory deficits. Inhibition of microglial NF-κB alone did not significantly impact spatial learning and memory, measured by the learning curve and the number of times to cross platform location in a 72 h probe trial (Fig. 5h, i). Strikingly, inactivating microglial NF-κB in PS19 mice significantly improved the learning ability (Fig. 5h) and restored the spatial memory in the probe trial (Fig. 5i), without affecting swimming speed (Fig. 5j). Thus, hyperactive microglial NF-κB plays a critical role in altering microglial homeostasis and driving cognitive deficits in PS19 mice.

**Inactivation of microglial NF-κB rescues tau-mediated transcriptomic changes and microglial autophagy deficits while increasing tau inclusions.** We next examined if the protective effects of microglial NF-κB inactivation are mediated by reducing tau inclusions. Surprisingly, instead of reducing neuronal tau inclusions, inactivation of microglial NF-κB markedly increase the number of intraneuronal tau inclusions in both hippocampus and cortex of 9–10-month-old PS19 mice (Fig. 6a–c). This indicates that the toxicities of neuronal tau inclusions on cognition might be regulated by NF-κB-dependent maladaptive microglia responses, which are reduced by NF-κB inhibition.

To dissect the mechanisms underlying the protective effects of inhibiting microglial NF-κB activity, we performed a bulk RNA-seq of cortical tissues from 8 to 9-month-old *Ikbkb^{+/+}, Ikbkb^{−/−}, Ikbkb^{+/+}; P301S+*, and *Ikbkb^{−/−}; P301S+* mice. Remarkably, the inactivation of microglial NF-κB resulted in a reversal of more than 90% of DEGs (286 out of 311 genes) in PS19 mice (Fig. 6d, e and Supplementary Data 13), despite elevating the tau inclusions. IPA analysis of these 286 genes revealed that the majority of reversed canonical pathways were related to inflammatory responses and the endo-lysosome system, and included neuroinflammation signaling, complement system, ROS generation, TLR signaling, autophagy, and phagocytosis (Fig. 6f and Supplementary Data 14). Tau-mediated biological function alterations including inflammatory response, cell movement, phagocytosis, superoxide production, and demyelination were also reversed (Supplementary Fig. 6a). The top upstream regulators of the pathways and functions reversed by NF-κB inactivation include those regulating immune cell survival and activation (e.g., *Tryobp, Spi1, Csf1r, Csf1*), superoxide generation (e.g., *Cybb*), and the interferon pathway (e.g., *Irf7, Irf8, Ifi16, Stat1*), suggesting these pathways are likely to be associated with the toxic effects of microglia in PS19 mice (Supplementary Fig. 6b). Our previous studies showed that impairments in chaperone-mediated autophagy (CMA) induced by acetylated tau lead to aberrant tau release[51], we therefore directly examined the effect of NF-κB inhibition on microglial CMA activity. We treated microglia expressing KFERQ-Dendra, a reporter for CMA[52], with tau fibrils in the presence or absence of NF-κB inhibitor. We found that NF-κB inhibition by TPCA-1 fully rescued the CMA deficit induced by tau fibrils, reflected by the recovery of the intracellular degradation of Dendra (Figs. 6g, h) and the amount of Dendra reporter reaching and internalized in lysosomes (Fig. 6g, i).

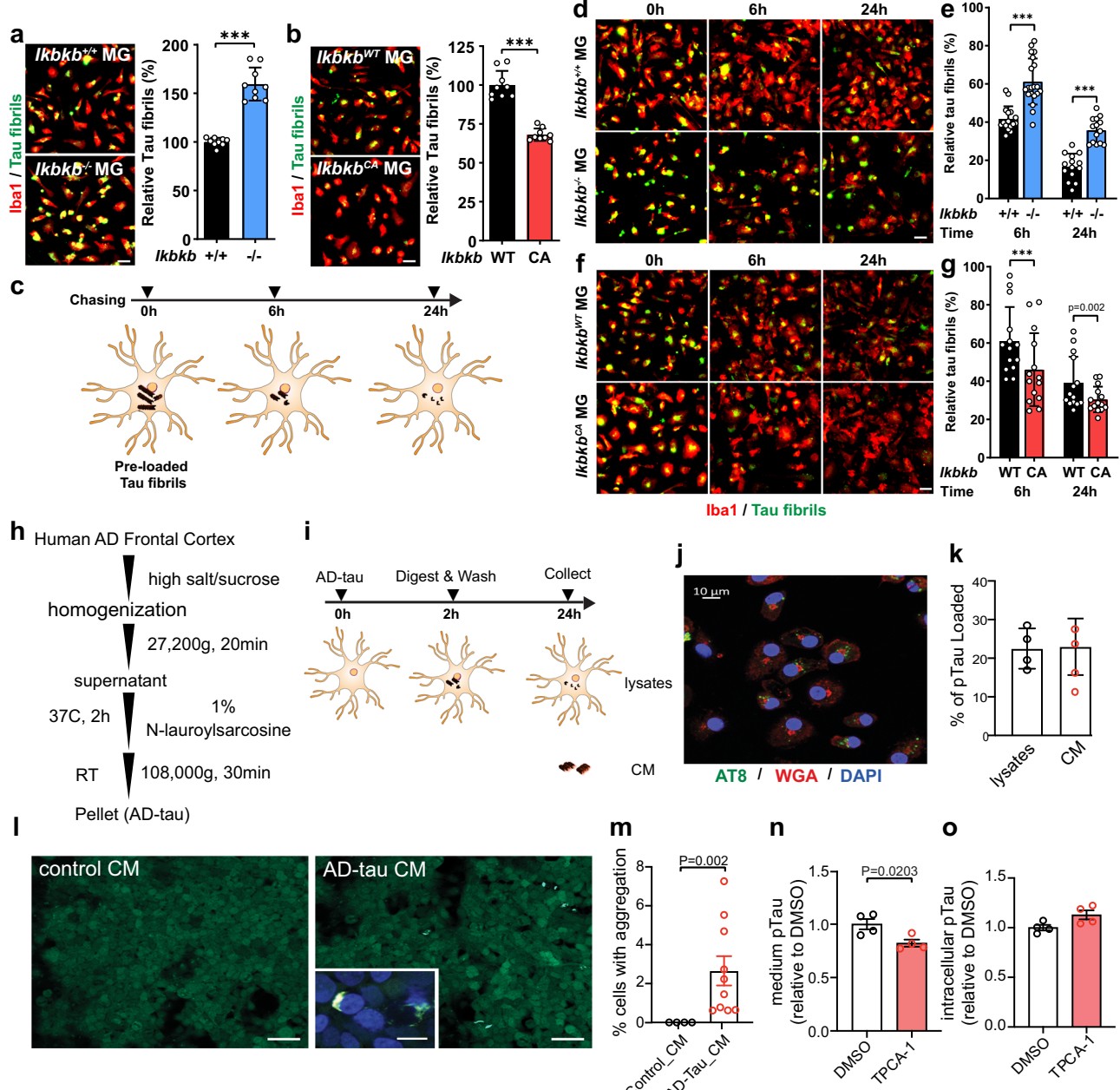

**Fig. 3 NF-κB promotes tau processing in primary microglia. a, b** Representative images and quantification of 24 h fluorescent tau fibrils accumulation in primary *Ikbkb*⁻/⁻ (**a**) and *Ikbkb*^CA (**b**) microglia compared with their littermate *Ikbkb*^+/+ and *Ikbkb*^WT control, respectively. Scale bar, 25 μm. Values are mean ± SD, relative to littermate control. Total *N* = 9 wells/group from three independent experiments. *P*-values were from multilevel mixed-effect model with experiment as hierarchical level, ***p < 0.001. **c** Schematic diagram of pulse-chase assay to quantify tau clearance in microglia. **d–g** Representative images and quantifications of pulse-chase assay comparing *Ikbkb*⁻/⁻ with *Ikbkb*^+/+ microglia (**d**, **e** 25,000 cells/well) and comparing *Ikbkb*^CA with *Ikbkb*^WT microglia (**f**, **g** 20,000 cells/well). Scale bar, 25 μm. Values are mean ± SD, relative to 0 h. Total *N* = 16 wells (6 h, *Ikbkb*^+/+); 20 wells(6 h, *Ikbkb*⁻/⁻); 14 wells/group of other conditions from three independent experiments. *P*-values were from multilevel mixed-effect model with experiment as hierarchical level, ***p < 0.001. **h** Schematic diagram illustrating the isolation procedures of human AD-tau. **i** Schematic diagram of quantifying release of AD-tau from microglia. **j** Detection of intracellular pTau by AT8 (green), cell membranes by Wheat Germ Agglutinin (WGA) (red), and nuclei by DAPI (blue). A representative image from three independent experiments. Scale bar: 10 μm. **k** Quantification of secreted and the intracellular pTau by AT8 ELISA and shown as the percentage of total pTau loaded in microglia after 2 h incubation. *N* = 4 wells. A representative dataset from two independent experiments. Values are mean ± SEM. **l** Representative images of tau aggregation in HEK biosensor cells induced by CM from microglia treated with AD-tau. Insert: High magnification confocal image of tau aggregation. Scale bar: 50 μm, insert: 15 μm. **m** Quantification of tau aggregation positive cells. *N* = 4 (control) and 10 (AD-tau) fields from three independent experiments, values are mean ± SEM, two-tailed Mann–Whitney test. **n, o** TPCA-1-treated microglia were incubated 2 h with AD-tau followed by 24 h chase. The pTau released to medium (**n**) and intracellular pTau (**o**) were determined by AT8 ELISA. A representative dataset from two independent experiments. *N* = 4 wells, values are mean ± SEM, relative to DMSO, two-tailed *t*-test. Source data are provided as a Source data file.

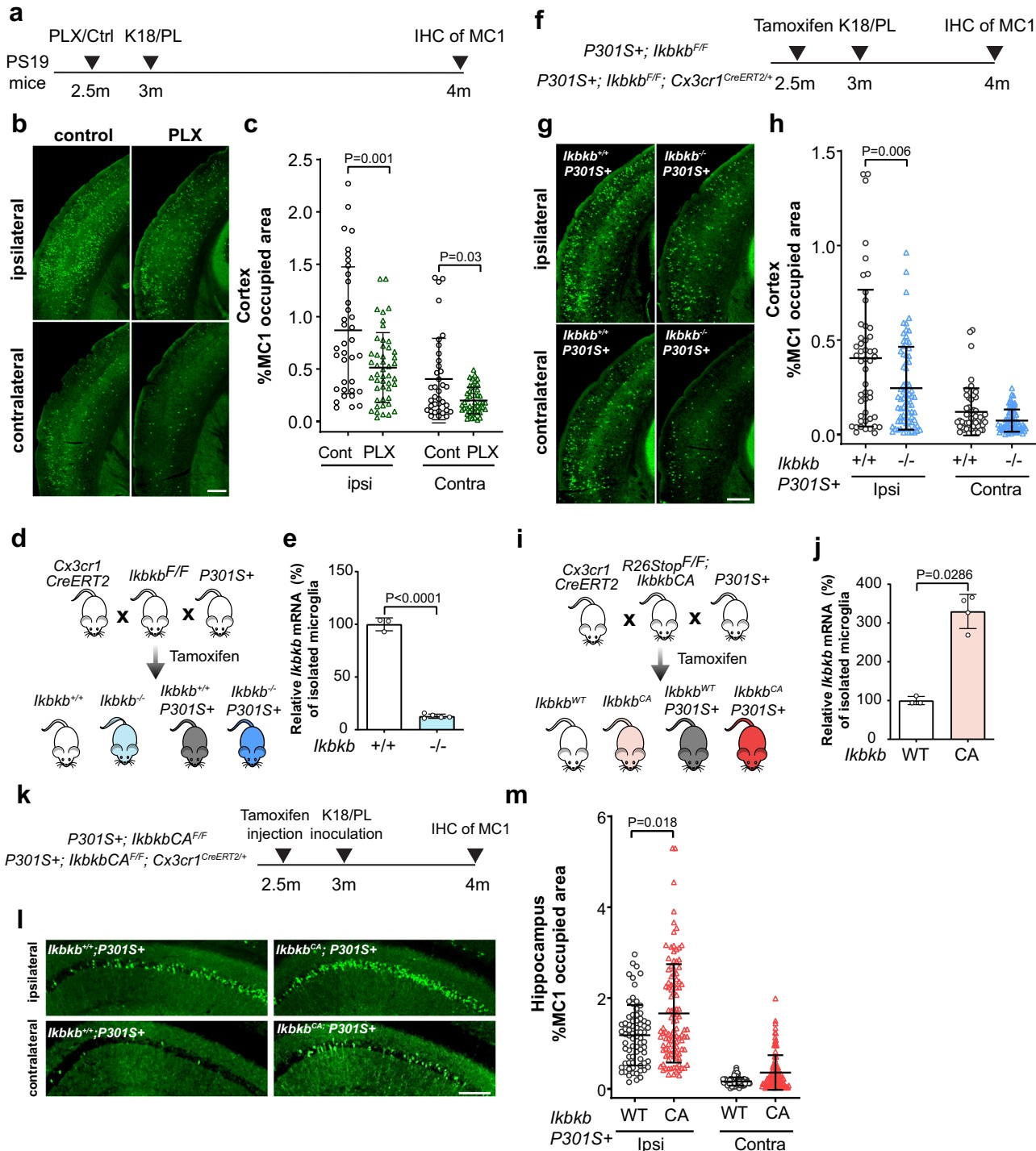

Increased association of reporter to lysosome resulted from primary activation of CMA as we did not find changes in the overall abundance of endo/lysosomal compartments quantified by LAMP1 (Figs. 6j, k). These results support that inhibiting microglial NF-κB prevents tau-induced cellular toxicities including autophagy deficits.

**NF-κB is required for tau-associated microglial states in PS19 mice.** Microglia exhibit disease-associated states in the presence of pathology[35], including tau[53], which have been characterized by single-cell RNA-seq. To characterize the effects of NF-κB on microglial transcriptomes, we performed single-nuclei RNA-seq (snRNA-seq)

using cortical tissues from *Ikbkb*[+/+]; *P301S+, Ikbkb*[−/−]; *P301S+*, and *Ikbkb*[CA]; *P301S+* mice. Non-transgenic (*Ikbkb*[+/+] and *Ikbkb*[WT], for simplicity, labeled as *Ikbkb*[+/+] in all snRNA-seq analysis) mice were used as non-disease controls. Following an established snRNA-seq protocol[54], we sequenced 114,118 nuclei from all four genotypes. After removal of potential multiplets using DoubletFinder[55] and filtering for low-quality nuclei determined by thresholding gene counts, UMI counts, and percentage mitochondrial genes per nuclei (Supplementary Fig. 7a–e), we used 103,681 nuclei for downstream analysis. Using reference gene markers for annotations (Supplementary Fig. 7g), we identified major cell types of the brain, which were similarly represented within each group and individual mouse (Fig. 7a, b, Supplementary Fig. 7f).

**Fig. 4 Microglial NF-κB activation promotes tau seeding and spread in PS19 mice. a** Schematic diagram illustrating the timeline of microglia depletion and K18/PL tau fibril inoculation in PS19 mice. Pathological tau was detected by MC1 immunohistochemical staining. **b, c** Representative images of MC1 tau positive neurons in ipsilateral and contralateral cortex (**b**) and quantification of percentage of MC1-occupied area (**c**) in control diet feeding mice ($n = 6$) and PLX diet feeding mice ($n = 8$), 6 sections per mice, values are mean ± SD. *P*-values were calculated using multilevel mixed-effect model with mouse as hierarchical level. Scale bar, 500 μm. **d** Breeding diagram illustrating the generation of PS19 mice with microglia-conditional deletion of IKKβ. **e** qPCR confirmed reduction of *Ikbkb* mRNA in adult microglia isolated from *Ikbkb*$^{-/-}$ mice ($n = 5$) compared with wild-type control *Ikbkb*$^{+/+}$ ($n = 3$) mice. Values are mean ± SD, two-tailed *t*-test. **f** Schematic diagram illustrating the timeline of microglial IKKβ deletion and K18/PL tau fibrils inoculation in PS19 mice. **g, h** Representative images of MC1 tau positive neurons in ipsilateral and contralateral cortex (**g**) and quantification of the percentage of MC1-occupied area (**h**) in *Ikbkb*$^{+/+}$; *P301S+* ($n = 10$) and *Ikbkb*$^{-/-}$; *P301S+* ($n = 15$)mice. 5 sections per mouse, values are mean ± SD. *P*-values were calculated using multilevel mixed-effect model with mouse as hierarchical level. Scale bar, 500 μm. **i** Breeding diagram illustrating the generation of PS19 mice with microglia expressing IKKβCA. **j** qPCR confirmed increase of *Ikbkb* mRNA in adult microglia isolated from *Ikbkb*$^{CA}$ mice ($n = 4$) compared with wild-type control *Ikbkb*$^{WT}$ mice ($n = 3$), values are mean ± SD, one-tailed Mann–Whitney test. **k** Schematic diagram illustrating the timeline of expressing microglial IKKβCA and K18/PL tau fibril inoculation in PS19 mice. **l, m** Representative images of MC1 tau positive neurons in ipsilateral and contralateral hippocampus CA1 region (**l**) and quantification of the percentage of MC1-occupied area (**m**) in *Ikbkb*$^{WT}$; *P301S+* ($n = 9$) and *Ikbkb*$^{CA}$; *P301S+* ($n = 14$) mice. 8 sections per mouse, values are mean ± SD, *P*-values were calculated using multilevel mixed-effect model with mouse as hierarchical level. Scale bar, 200 μm. Source data are provided as a Source data file.

We then examined the trajectory of the subclusters of a total 4305 microglia (1068 from *Ikbkb*$^{+/+}$, 1275 from *Ikbkb*$^{+/+}$; *P301S+*, 720 from *Ikbkb*$^{-/-}$; *P301S+*, and 1242 from *Ikbkb*$^{CA}$; *P301S+*) to investigate how NF-κB affects microglial states in P301S mice using Monocle[56]. The microglial population from the four genotypes exhibited a clearly-defined five subclusters (Fig. 7c, Supplementary Data 15). Analyses of the distribution of the five subclusters by genotypes revealed that microglia from *Ikbkb*$^{+/+}$ brains were distributed among clusters 1 and 2, while the vast majority of cluster 3 came from *Ikbkb*$^{+/+}$; *P301S*, likely representing tau-induced microglial states (Fig. 7d). Most microglial from clusters 4 and 5 came from *Ikbkb*$^{CA}$; *P301S*, suggesting that constitutive NF-κB activation further transforms microglial states (Fig. 7d). In direct contrast, microglia from *Ikbkb*$^{-/-}$; *P301S* were mostly found in Cluster 1, along with wild-type *Ikbkb*$^{+/+}$ microglia, demonstrating that activation of NF-κB is required for tau-induced transcriptomal changes (Fig. 7d). We next examined microglial trajectory with pseudotime from 0 to 15 (Fig. 7e). Microglia from non-transgenic control (*Ikbkb*$^{+/+}$) mice were enriched at the starting point of the trajectory (pink, Fig. 7e, f). Microglia from PS19 mice with wild-type IKKβ (*Ikbkb*$^{+/+}$; *P301S+*) exhibit trajectory away from those of non-transgenic control (green, Fig. 7f), while those lacking IKKβ (*Ikbkb*$^{-/-}$; *P301S+*) partially overlaps with that of *Ikbkb*$^{+/+}$ control, illustrating their failure to advance in the direction of that of PS19 microglia (purple, Fig. 7e). The other end of the trajectory (yellow, Fig. 7e) was populated with microglia from *Ikbkb*$^{CA}$; *P301S+* mice, extending beyond the tau-associated disease states (cyan, Fig. 7f). Thus, tau-associated disease states in microglia require NF-κB activation.

We further dissected the pathways modified by microglial NF-κB in PS19 mice. We identified 961 DEGs in *Ikbkb*$^{-/-}$; *P301S+* microglia and 691 DEGs in *Ikbkb*$^{CA}$; *P301S+* microglia (vs. *Ikbkb*$^{+/+}$; *P301S+*) (FDR < 0.05, |log2FC| >0.1, Supplementary Fig. 8a, b, Supplementary Data 16). We found that genes induced by tau positively correlated with DAM genes (Supplementary Fig. 8c), while those induced by *Ikbkb*$^{CA}$ exhibited no correlation, indicating distinct microglial states (Supplementary Fig. 8d). Indeed, the DEGs upregulated in tau-induced cluster 3 vs. cluster 1 (*Ikbkb*$^{-/-}$; *P301S+* and *Ikbkb*$^{+/+}$) partially overlap with those upregulated by constitutive NF-κB activation, with 439 DEGs uniquely upregulated in *Ikbkb*$^{CA}$; *P301S+* microglia (vs. *Ikbkb*$^{+/+}$; *P301S+*) (FDR < 0.05, |log2FC| >0.1, Fig. 7g, Supplementary Data 17). Given the protective effects of NF-κB inactivation against tau-mediated toxicity, we were particularly interested in the 200 DEGs downregulated by NF-κB inactivation by comparing gene expression in cluster 1 vs. cluster 3 (Fig. 7h), which could underlie its protective effects against functional deficits and tau seeding/spread. Investigation of the overlapping pathways downregulated by NF-κB inactivation using gene set enrichment analysis (GSEA) identified complement, the IL2/STAT5, and lipid binding pathways (Fig. 7h, Supplementary Fig. 8a). Consistent with our finding that microglial NF-κB promotes tau processing, release, seeding, and spread, both proteolysis and exocytosis functions of microglia were downregulated in *Ikbkb*$^{-/-}$; *P301S+* microglia and upregulated in *Ikbkb*$^{CA}$; *P301S+* microglia by comparing gene expression in clusters 4 and 5 vs. cluster 1 (Fig. 7i). These pathways include secretory granule complex, vesicular/protein transport, and exocytosis, trafficking (Fig. 7h, i). The reprogramming of microglial disease states by NF-κB inactivation provides molecular underpinnings for microglial-mediated tau seeding and tau-mediated cognitive deficits.

## Discussion

Here, we show that microglial NF-κB activation is required for microglial-mediated tau spreading and tau-mediated spatial learning and memory deficits in tauopathy mice (Fig. 8). NF-κB signaling was among the top altered cellular immune response pathways in response to tau in microglia isolated from PS19 tauopathy mice. By genetically activating or inactivating microglial NF-κB, we found that NF-κB accelerates microglial processing of tau in cultured microglia, and promotes tau seeding and spreading in vivo. Moreover, inactivation of NF-κB in tauopathy mice partially restored microglial homeostasis, reversed tau-mediated changes in the transcriptome and the CMA function, and rescued spatial learning and memory deficits. By identifying tau-associated microglial states that were diminished by NF-κB inactivation, our snRNA-seq analyses further reveal potential molecular mechanisms underlying microglial-mediated tau seeding/spreading and toxicity.

In both tau-stimulated primary microglia and isolated microglia from aged PS19 mice, we found that NF-κB is among the top upstream regulators, consistent with a previous study of microglia from rTg4510 mice[57]. We also analyzed microglial DEGs from another study of Tau-P301S model (GSE93180[58], Supplementary Data 18) and confirmed the activation of NF-κB pathway (Supplementary Fig. 9). In contrast, NF-κB target genes were not enriched in a microglial transcriptome study from APPswe/PS1dE9 mice, an Aβ driven AD model[59], suggesting distinct disease-associated transcriptional programs induced by tau vs. Aβ. Microglial NF-κB could be activated by soluble tau monomers, in agreement with a previous study showing that microglial NF-κB is activated as early as 2 months of age in the rTg4510 model[57]. We observed that some of the pathways and functions

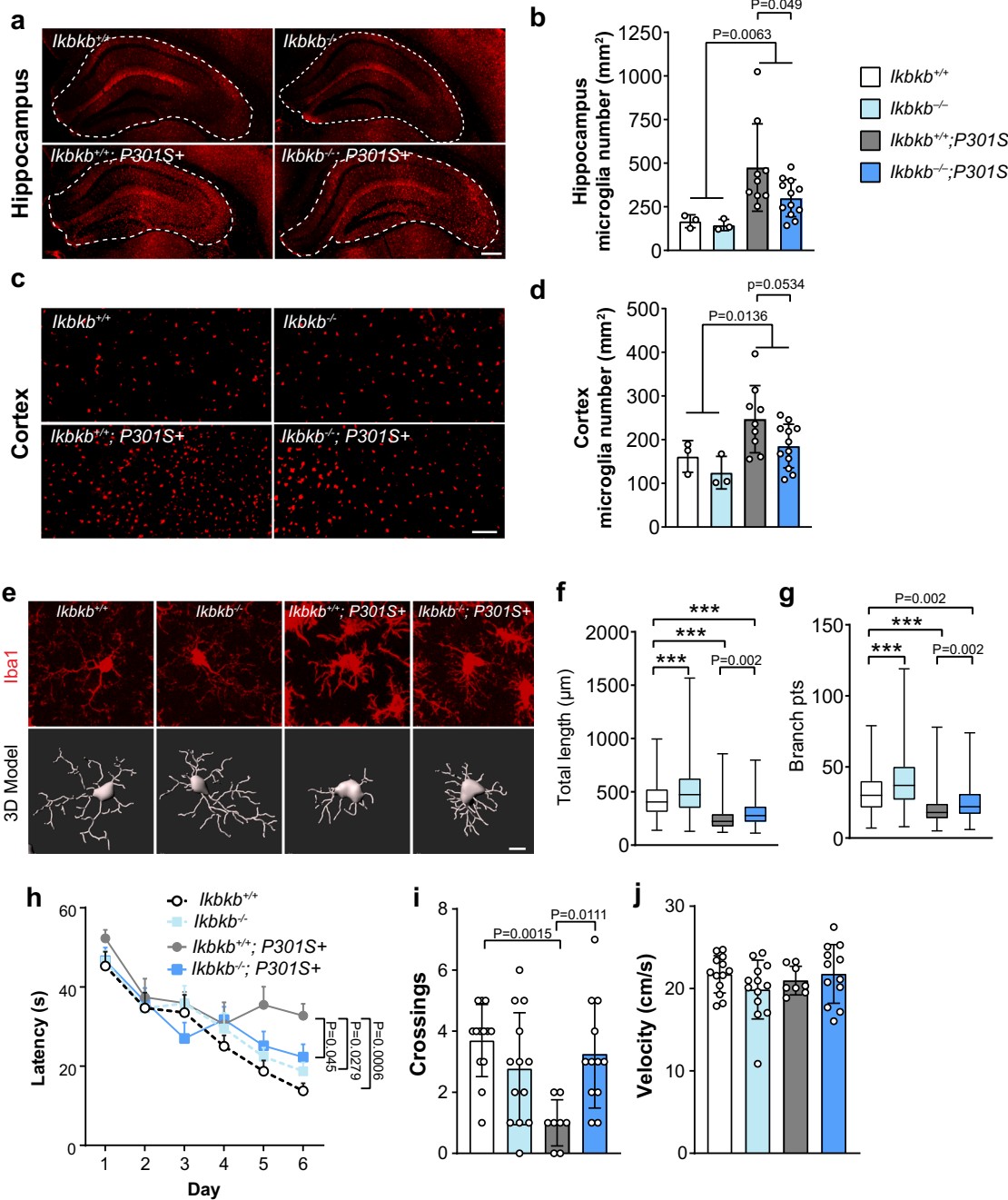

**Fig. 5 Inactivation of microglial NF-κB partially rescues microgliosis, morphological alterations and protects against spatial learning and memory deficits in PS19 mice. a–d** Representative immunohistochemical staining (**a**, **c**) and quantification (**b**, **d**) of Iba1+ microglia in the hippocampus (**a**, **b**) and cortex (**c**, **d**) of *Ikbkb*$^{+/+}$ ($n=3$), *Ikbkb*$^{-/-}$ ($n=3$), *Ikbkb*$^{+/+}$; *P301S+* ($n=9$), and *Ikbkb*$^{-/-}$; *P301S+* ($n=12$) mice. Scale bar, hippocampus 250 μm, cortex 100 μm, values are mean ± SD, two-way ANOVA with Sidak's multiple comparisons post-test. **e** Representative immunohistochemical confocal images showing morphological features of microglia from the hippocampus of *Ikbkb*$^{+/+}$, *Ikbkb*$^{-/-}$, *Ikbkb*$^{+/+}$; *P301S+*, and *Ikbkb*$^{-/-}$; *P301S+* mice and the corresponding 3D reconstructions using Imaris. Scale bar, 10 μm. **f**, **g** Quantification of the total length of microglial processes (**f**) and the number of microglia process branch points (**g**). $N=4$ mice per genotype (four sections per mouse) were imaged and a total of 893 (*Ikbkb*$^{+/+}$), 902 (*Ikbkb*$^{-/-}$), 3455 (*Ikbkb*$^{+/+}$; *P301S+*), and 2143 (*Ikbkb*$^{-/-}$; *P301S+*) microglia were analyzed. Values are shown as boxplot. The box extends from the 25th to 75th percentiles with the median is shown in the middle of box. Whiskers show from the smallest value to the largest. *P*-values were calculated using multilevel mixed-effect model with mouse and section as hierarchical levels, ***$P < 0.001$. **h–j** Morris Water Maze test was performed using *Ikbkb*$^{+/+}$ ($n=13$), *Ikbkb*$^{-/-}$ ($n=13$), *Ikbkb*$^{+/+}$; *P301S+* ($n=8$), and *Ikbkb*$^{-/-}$; *P301S+* ($n=12$) mice. **h** Escape latency was plotted against the training days; **i** Times of crossing the platform location in the 72 h probe trial; **j** swimming speed. Values are mean ± SD, two-way ANOVA with Tukey's multiple comparisons post-test. Source data are provided as a Source data file.

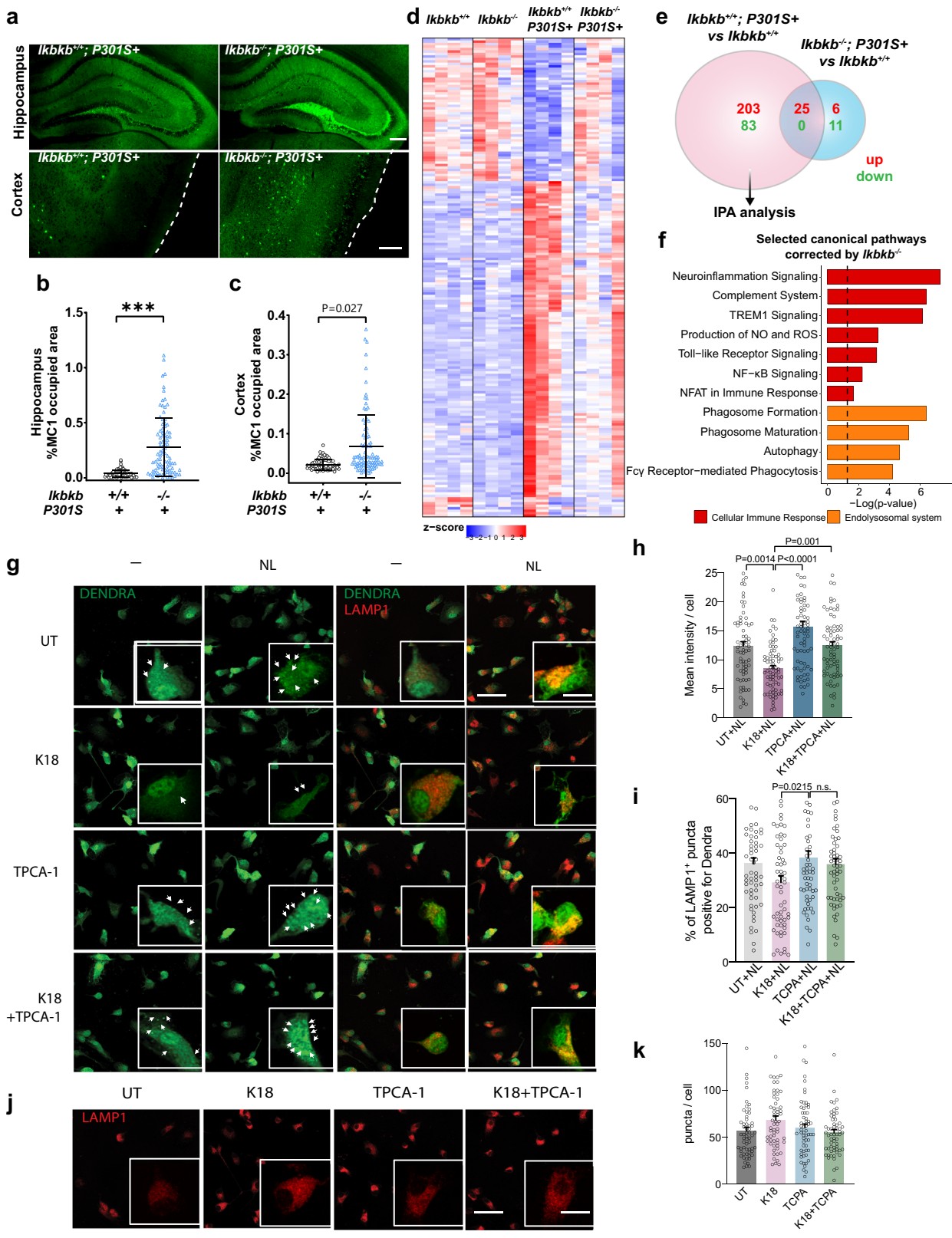

induced by tau overlapped with those induced by NF-κB activation, including proliferation and cell movement/migration, and that these could be reversed by inactivating microglial NF-κB in vitro and in PS19 mice. Moreover, inhibition of microglial NF-κB activity in PS19 mice reduced microgliosis, and resulted in longer and more branchy processes, partially reverting cells to a

more homeostatic microglial state. Our snRNA-seq analyses further revealed that NF-κB is required for tau-associated microglial states in PS19 mice. NF-κB inactivation diminished the tau-associated disease states. In contrast, constitutive NF-κB activation in PS19 mice further extended microglial states beyond the tau-associated disease states, supporting a feed-forward

**Fig. 6 Inactivation of microglial NF-κB rescues tau-induced transcriptomic changes and microglial autophagy deficits while increasing tau inclusions.**
**a** Representative images of MC1 tau in the hippocampus and cortex of *Ikbkb*[+/+]; *P301S*+ and *Ikbkb*[−/−]; *P301S*+ mice. Scale bar, 250 μm **b**, **c** Quantification of the percentage of MC1-occupied area in the hippocampus (**b**) and cortex (**c**) of *Ikbkb*[+/+]; *P301S*+ (*n* = 9) mice and *Ikbkb*[−/−]; *P301S*+ (*n* = 12) mice; 8 sections/mouse. *P*-values were calculated using multilevel mixed-effect model with mouse as hierarchical level, ***P < 0.001. **d–h** Bulk RNA-seq analysis of cortical tissues from *Ikbkb*[+/+], *Ikbkb*[−/−], *Ikbkb*[+/+]; *P301S*+, and *Ikbkb*[−/−]; *P301S*+ mice (*n* = 4 mice/genotype). DEGs were generated in comparison to *Ikbkb*[+/+] mice. **d** Heatmap representing DEGs across all genotypes. Normalized and z-scored (color-coded) value of log₂PFKM of each gene is shown. Shades of red: upregulation; shades of blue: downregulation. DEGs and biological replicate samples were hierarchically clustered. **e** Venn diagram comparing the overlap of DEGs in *Ikbkb*[+/+]; *P301S*+ and *Ikbkb*[−/−]; *P301S*+ mice. **f** Selected IPA canonical pathways identified for 286 DEGs corrected by microglial IKKβ deletion in PS19 mice as in **e**. *P*-values were calculated using right-tailed Fisher's exact test with threshold of significant enrichment as *p*-value ≤ 0.05 (indicated by a dotted line of −log (*p*-value) = 1.3). **g–k** Primary microglia from KFERQ-Dendra mice were incubated with K18/PL tau fibrils with or without TPCA-1. Lysosomal degradation of the CMA reporter was determined by inhibiting lysosomal proteases with NH₄Cl and leupeptin (N/L). **g** Representative images of KFERQ-Dendra microglia co-stained with LAMP1. Left: single channel for Dendra; right: merged channels. Scale bar, 50 μm. Insets: 10 μm. Arrows: Dendra + puncta. UT untreated control. **h** Quantification of Dendra intensity in microglia incubated with N/L as in **g**. **i** Percentage of total LAMP1 puncta positive for Dendra signal in microglia incubated with N/L as in **g**. **j** Representative LAMP1 immunostaining in microglia incubated with K18/PL tau fibrils with or without TPCA-1. Scale bar, 50 μm. Insets, 10 μm. **k** Quantification of average number of LAMP1 positive puncta as in **j**. **h** 72; **i** and **k** 60 cells from 3 independent experiments, values are mean ± SEM. One-way ANOVA with Tukey post-hoc test. Source data are provided as a Source data file.

mechanism between tau toxicity and microglial NF-κB activation (Fig. 8).

Emerging evidence supports the hypothesis that microglia participate in tau seeding and spreading[14,17]. NLRP3–ASC inflammasome activation was found to exacerbate exogenously seeded tau pathology as well as non-exogenously seeded intraneuronal tau aggregates, at least partially through modulating tau phosphorylation[60,61]. Our study shows that in PS19 mice, exogenous tau inoculation-induced tau seeding was accelerated by microglial NF-κB activation, but diminished by NF-κB inactivation. Indeed, inactivation of microglial NF-κB signaling alone induced a similar reduction in seeding as depleting microglia altogether, highlighting the central role of NF-κB signaling in microglia-mediated tau seeding. NF-κB activation could promote microglia to secrete more seeding-competent tau, and thus accelerate the spread of tau pathology. Indeed, in cultured microglia, we showed that NF-κB inhibition rescued deficits in CMA and reduced tau secretion, consistent with our previous finding that impairments in CMA promote extracellular tau release in neurons[51].

How NF-κB signaling mediates the opposite effects on intraneuronal tau inclusions triggered by exogenous tau seeds vs. those induced by transgene alone is not known. These two models develop tau inclusions at vastly different timeline: 1-month post-inoculation vs. 8–9 months, and likely involve distinctive proteostasis mechanisms. Beyond modulating tau phosphorylation[16], microglial NF-κB may also modulate the sorting, trafficking, and exocytosis of tau in neurons. Further investigation is needed to determine the extracellular soluble and seeding-competent tau released from microglia, and to identify the cellular machinery involved in sorting, transportation, degradation, and release of tau in both neurons and glia.

Microglia activation in neurodegenerative diseases can have both beneficial and detrimental effects. Similarly, in response to tauopathy, some of the microglial responses are adaptive, serving protective functions, while others could be maladaptive and promote toxicity. Inhibiting microglial NF-κB activity is sufficient to rescue the learning and memory deficits in PS19 mice, suggesting that NF-κB hyperactivation drives tau toxicity in neurons. Single-nuclei RNA-seq analyses of transcriptomes of excitatory and inhibitory neurons revealed that inactivation of microglial NF-κB caused marked changes in these neurons (Supplementary Fig. 10). Microglial NF-κB inactivation led to elevated synaptogenesis pathways in excitatory neurons, and endocannabinoid synapse and P2Y purigenic receptor signaling pathways in inhibitory neurons, consistent with the protective effects of inactivating NF-κB. In transcriptomic analyses of tau-stimulated microglia in culture and in vivo, we identified that elevation of cytokines such as IL-1, IL-6, TNFα, and

interferons is among the major consequences of activation of microglial NF-κB. Chronic elevation of these cytokines can cause neurotoxicity. In other models of neurodegeneration, inhibition of microglial NF-κB reduced inflammatory markers, rescued motor neuron death, and extended survival of ALS mice[62]. In a kainic acid-induced neurodegenerative mouse model, deletion of IKKβ in microglia reduced expression of IL-1β and TNFα, which resulted in 30% reduction of hippocampal neuronal cell death[63]. Another potential mechanism mediating the toxic maladaptive responses could be the elevated expression of the complement system and related genes (e.g., *C1qa, C1qc*), which are also positively regulated by microglial NF-κB[64,65]. In AD mouse models, activated microglia were found to phagocytose synapses, a process relying on the activation of complement factors, including C3, C1q, and CR3[66,67]. In tauopathy, inactivation of C3-C3aR signaling reverses the deregulation of the immune network and rescues behavior deficits in PS19 mice[68]. The exact protective mechanism underlying inactivating microglial NF-κB remains to be determined.

Using two in vivo models, our current study revealed important roles of microglial NF-κB activity in tau spreading as well as tau-induced neuronal toxicity and cognitive deficits (Fig. 8). In young PS19 mouse, the spreading of tau inclusion is driven by exogenous tau seeds in a rather short period of time. We showed that microglial NF-κB activation accelerates while deficiency reduces the spread of tau inclusions. Combined with our finding that NF-κB inhibition reduced the release of tau from cultured primary microglia, our results suggest that microglial NF-κB activation promotes tau spreading via enhanced release of seedable tau (Fig. 8a). In contrast, in the non-seeded PS19 model, age-dependent tau aggregation is determined by tau proteostasis of neurons expressing human tau transgene. In this chronic model, we found that the protective effects of microglial NF-κB inhibition are associated with normalization of >90% of the overall transcriptome, abolishment of tau-mediated microglial states at the single-cell level, and elevation of intraneuronal tau inclusions. While the mechanisms underlying how microglial NF-κB affects tau proteostasis in neurons are unknown, it is possible that microglial NF-κB activity may stimulate tau exocytosis and/or proteolysis in these neurons, resulting in elevated intraneuronal tau inclusions in brains lacking microglial NF-κB activity (Fig. 8b).

Our surprising finding that inactivating microglial NF-κB protected against tau-mediated cognitive deficits despite elevated tau inclusions reveals that tau pathology and toxicity are multi-faceted and cell-type-specific. While intraneuronal tau load can be pathological, it does not represent all pathological tau species, such as those secreted that can exert toxicity as well. Thus, the amount of intraneuronal tau inclusions may not directly correlate with the extent of neuronal injury. Importantly, microglial NF-κB

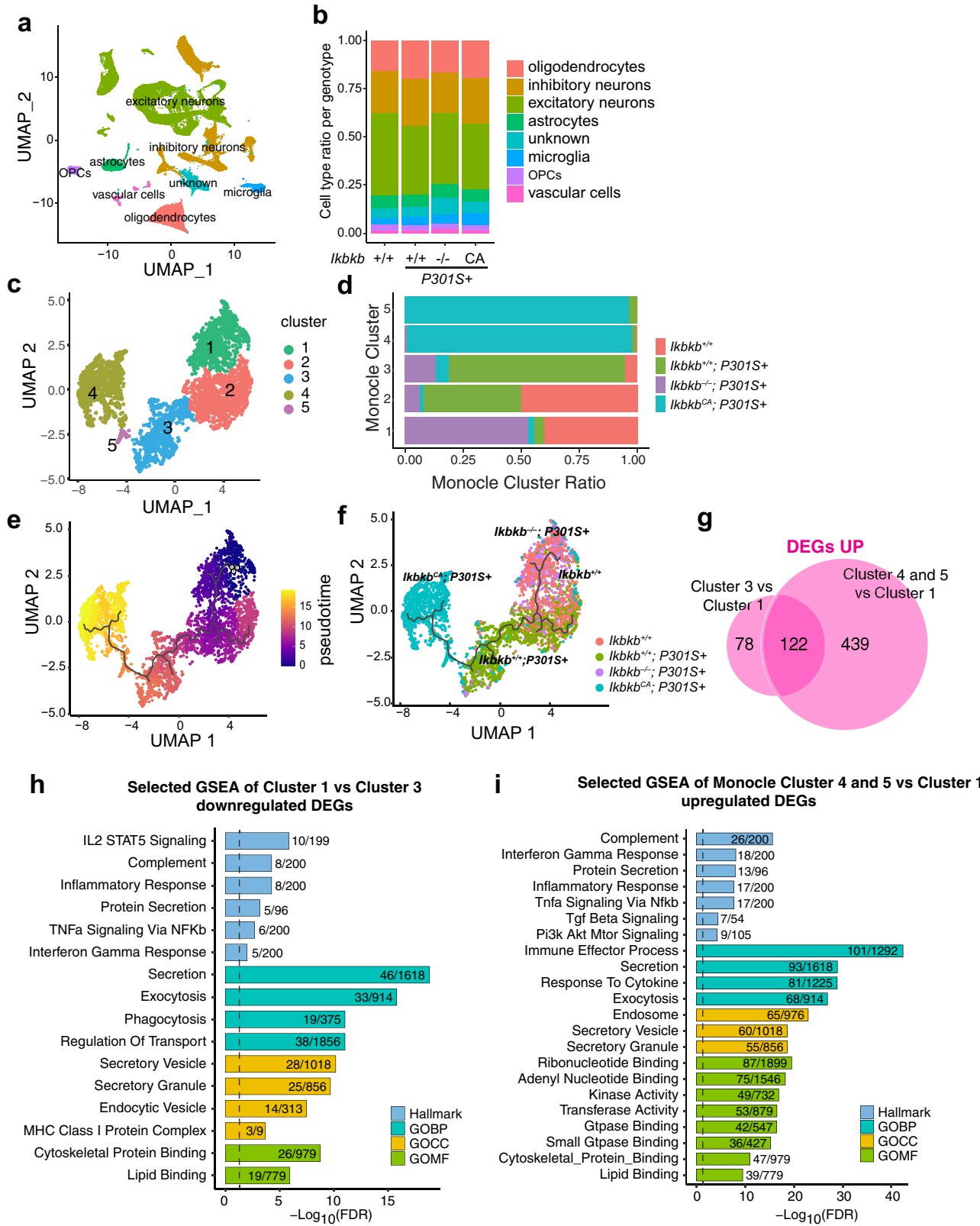

activity is necessary for executing tau-mediated neuronal toxicity and cognitive deficits. Taken together, our work shows that microglial NF-κB acts downstream of tau pathology, and directly mediates toxic effects on cognition, highlighting the potential of blocking maladaptive microglial responses instead of removing tau aggregates as a therapeutic strategy to treat tauopathy.

## Methods

**Mice**. Tau-P301S transgenic mice (P301S+,JAX:008169) or *Cx3cr1*<sup>CreERT2/ CreERT2</sup> mice (JAX: 021160) were crossed with *Ikbkb*<sup>F/F</sup> mice (MGI:2445462) or R26-Stop<sup>FL</sup>ikk2ca (*ikbkbCA*<sup>F/F</sup>) mice (JAX:008242) to obtain *P301S+;Ikbkb*<sup>F/F</sup>, *P301S+; Ikbkb*<sup>CA/F/F</sup>, *Cx3cr1*<sup>CreERT2/+</sup>; *Ikbkb*<sup>F/F</sup>, and *Cx3cr1*<sup>CreERT2/+</sup>; *IkbkbCA*<sup>F/F</sup> mice. *P301S+;Ikbkb*<sup>F/F</sup> mice were then crossed with *Cx3cr1*<sup>CreERT2/+</sup>; *Ikbkb*<sup>F/F</sup> mice to obtain *P301S+;Cx3cr1*<sup>CreERT2/+</sup>; *Ikbkb*<sup>F/F</sup> mice and littermates controls including *Ikbkb*<sup>F/F</sup>, *Cx3cr1*<sup>CreERT2/+</sup>; *Ikbkb*<sup>F/F</sup>, and *P301S+;Ikbkb*<sup>F/F</sup> mice. Similarly, *P301S+;IkbkbCA*<sup>F/F</sup>

**Fig. 7 NF-κB is required to induce tau-associated microglia states in PS19 mice.** Single-nuclei RNA-seq were performed using cortical tissues from *Ikbkb*$^{+/+}$ (*n* = 4), *Ikbkb*$^{+/+}$; *P301S*+ (*n* = 3), *Ikbkb*$^{-/-}$; *P301S*+ (*n* = 2), and *Ikbkb*$^{CA}$; *P301S*+ (*n* = 2) mice. **a** UMAP plots of all single nuclei and their annotated cell types. OPCs Oligodendrocyte progenitor cells. **b** Proportion of cell types for each genotype. **c** UMAP plot depicting different microglial cell subclusters. Each cell was color-coded based on their cluster affiliation. **d** Proportion of cells in each microglial subcluster for each genotype. **e** Pseudotime-time trajectory demonstrating the shift of microglial state. **f** Pseudotime-time trajectory of microglia labeled with genotypes. The trajectory illustrates the shift of microglial transcriptome from *Ikbkb*$^{-/-}$; *P301S*+ to *Ikbkb*$^{CA}$; *P301S*+ mice. **g** Venn diagram comparing the number and overlap of upregulated DEGs in Cluster 3 vs Cluster 1, and Cluster 4 and 5 vs Cluster 1. **h, i** Selected GSEA hallmark and Gene Ontology pathways identified for downregulated microglia DEGs in Cluster 1 vs Cluster 3 (**h**) and upregulated microglia DEGs in Cluster 4 and 5 vs Cluster 1 (**i**). GOBP GO biological processes, GOCC GO cellular compartment, GOMF GO molecular function.

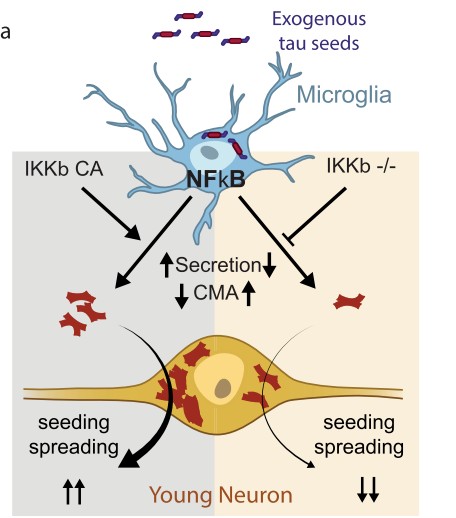
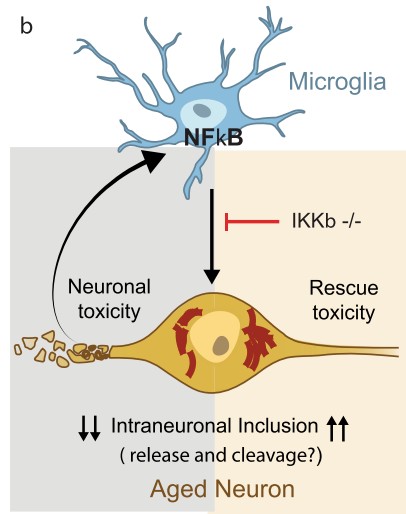

**Fig. 8 Hypothetical model of microglia NF- κB drives tau seeding/spreading and toxicity. a** In an acute model of young PS19 mouse, microglia process the inoculated exogenous tau and release seedable tau to drive tau spreading in a month. Microglial NF-κB activation accelerates the spread of tau inclusions seeded by the inoculation of exogenous tau seeds, likely due to increased tau secretion and impaired autophagy, which is rescued by NF-κB inhibition. **b** In the non-seeded PS19 model, tau aggregation in aged neurons is determined by tau proteostasis of neurons expressing human mutant tau transgene. Microglial NF-κB activation drives while deletion rescues tau toxicities. NF-κB activity drives down intraneuronal inclusions via unknown mechanisms, likely via stimulating tau release and/or proteolytic cleavage in these neurons.

mice were then crossed with *Cx3cr1*$^{CreERT2/+}$; *IkbkbCA*$^{F/F}$ mice to obtain *P301S*+;*Cx3cr1*$^{CreERT2/+}$; *IkbkbCA*$^{F/F}$ mice and littermates controls including *IkbkbCA*$^{F/F}$, *Cx3cr1*$^{CreERT2/+}$; *IkbkbCA*$^{F/F}$, and *P301S*+; *IkbkbCA*$^{F/F}$ mice. Mice were housed no more than 5 per cage, given ad libitum access to food and water, and were housed in a pathogen-free barrier facility with 72 °F, 50% humidity, 12-h light on/off cycle. Mice of both sexes were used for all experiments except for RNA-seq, which used only male mice. All animal work was performed in accordance with NIH guidelines and protocols approved by University of California, San Francisco, Institutional Animal Care and Use Committee.

**Tamoxifen administration.** To induce efficient Cre expression and recombination in vivo, tamoxifen (T5648, Sigma–Aldrich) was dissolved in corn oil to prepare 20 mg/ml stock. Mice were given tamoxifen via intraperitoneal (IP) injection at 2 mg per day for 10 days between 2.5 months to 3 months of age. To induce Cre expression and recombination in primary microglia, 4-hydroxytamoxifen (SML1666, Sigma–Aldrich) were incubated with glia mixed culture for 3 days at 5ug/ml during day 10–12 of culture. Microglia were centrifuged to completely remove 4- hydroxytamoxifen before use. Littermate control mice or microglia that do not have Cre gene were also administrated tamoxifen. Gene symbols, corresponding genotype, and IKKβ levels in microglia are summarized as follows.

**Primary microglia culture.** Primary microglia were prepared as described previously[69]. Briefly, mouse hippocampi and cortices from 2–3-day-old newborn pups were isolated in DPBS. After removing the meninges, brain tissues were cut into small pieces and digested with 0.1% trypsin at 37 °C for 20 min before neutralizing the trypsin with 30% FBS/DMEM media. Digested tissues were triturated to the cell suspension and centrifuged for 15 min at 200 × *g*. After resuspension in 10% FBS/DMEM, cells were plated onto poly-D-lysine (PDL)-coated T-75 flasks to generate mixed glial cultures in the medium of 10% FBS/DMEM. When confluent on day 12, microglia were separated from the glia layer by shaking the flask at 200 rpm for 3 h. Floating microglia were collected for RNA analysis or seeded at 75,000 cells/cm² in PDL-coated plates for tau fibril stimulation and processing assay. For microglial tau release assay, 5 ng/ml GM-CSF was added for microglial culture but removed after cells were harvested. Then microglia were seeded on culture plates in the medium of 10%FBS/DMEM without GM-CSF for 24 h before treatment.

**Adult microglia isolation.** Adult microglia were isolated using magnetic-activated cell sorting (MACS) as described before[70]. Briefly, anesthetized mice were thoroughly transcardially perfused with cold PBS to remove circulating blood cells in the CNS. Dissected brains were chilled on ice and minced in digestion media containing 0.2% collagenase type 3 (LS004182, Worthington) and 3 U/mL dispase (LS02104, Worthington). After 37 °C incubation for 45 min, digestion was inactivated by 2.5 mM EDTA (15575020, Thermofisher) and 1% fetal bovine serum (10082147, Thermofisher). Digested brain tissues were triturated by serological pipette to the cell suspension and passed through a 70 μm cell strainer. Myelin in the cell suspension was depleted by myelin removal beads (130-096-733, Miltenyi Biotec) and magnetic LD columns (130-042-901, Miltenyi Biotec). Adult microglia were finally enriched from the eluant by CD11b MicroBeads (130-049-601, Miltenyi Biotec) and magnetic MS column (130-042-201, Miltenyi Biotec) for RNA isolation.

| Gene symbols | Genotype | IKKβ levels in microglia |
| --- | --- | --- |
| *Ikbkb*$^{+/+}$ | *Cx3cr1*$^{+/+}$; *Ikbkb*$^{F/F}$ | Normal |
| *Ikbkb*$^{-/-}$ | *Cx3cr1*$^{CreERT2/+}$; *Ikbkb*$^{F/F}$ | Deleted |
| *Ikbkb*$^{+/+}$; *P301S*+ | *P301S*+;*Cx3cr1*$^{+/+}$; *Ikbkb*$^{F/F}$ | Normal |
| *Ikbkb*$^{-/-}$; *P301S*+ | *P301S*+;*Cx3cr1*$^{CreERT2/+}$; *Ikbkb*$^{F/F}$ | Deleted |
| *Ikbkb*$^{WT}$ | *Cx3cr1*$^{+/+}$; *IkbkbCA*$^{F/F}$ | Normal |
| *Ikbkb*$^{CA}$ | *Cx3cr1*$^{CreERT2/+}$; *IkbkbCA*$^{F/F}$ | High |
| *Ikbkb*$^{WT}$; *P301S*+ | *P301S*+; *Cx3cr1*$^{+/+}$; *IkbkbCA*$^{F/F}$ | Normal |
| *Ikbkb*$^{CA}$; *P301S*+ | *P301S*+; *Cx3cr1*$^{CreERT2/+}$; *IkbkbCA*$^{F/F}$ | High |

**RNA isolation and quantitative PCR.** To quantify target gene expression, RNA from primary microglia, isolated microglia, and cortical tissue were extracted by Direct-zol RNA micro-prep kit (R2061, Zymo Research) following the manufacturer's instructions. Genomic DNA was digested by DNase I. For qPCR, RNA was reverse-transcribed to cDNA by iScript cDNA synthesis kit (1708890, Bio-Rad). The qPCR reactions were performed using SYBR Green PCR Master Mix (4309155, Applied Biosystems) on ABI 7900HT real-time system (Applied Biosystems). GAPDH was used as a reference gene for normalization and the relative expression differences were calculated based on the $2^{-\Delta\Delta Ct}$ method. The following primers were used for RT-qPCR: *Gapdh* (forward) 5′-GGGAAGCCCATCAC-CATCTT-3′, (reverse) 5′-GCCTTCTCCATGGTGGTGAA-3p; *Ikbkb*: (forward) 5′-AAGAACAGAGACCGCTGGTG -3′ and (reverse) 5′- CAGGTTCTG-CATCCCCTCTG -3′.

**Brain tissue collection.** Mice were anesthetized with tribromoethanol and transcardially perfused with PBS. Hippocampus and cortex were dissected from one hemibrain and flash-frozen at −80 °C for biochemical analyses and snRNA-seq, whereas the other half hemibrain (or whole brain from tau spreading experiment) was fixed in 4% paraformaldehyde for 48 h, followed by 30% sucrose infiltration for 48 h at 4 °C. 30 μm-thick coronal brain sections were prepared by freezing microtome (Leica) and stored at −20 °C in cryoprotectant before staining.

**Immunohistochemistry and image analysis.** DPBS was used for immunohistochemistry. Six to eight pieces of brain sections per mouse that contain a series of anterior to posterior hippocampus were washed to remove cryoprotectant and then permeabilized by 0.5% Triton X-100. After blocking in 5% normal goat serum (NGS) for 1 h, brain sections were incubated with primary antibodies in the same blocking buffer overnight at 4 °C. Sections were then washed by DPBS containing 0.1% Tween-20 and incubated with Alexa-conjugated secondary antibodies (1:500) for 1 h in blocking buffer. After washing, sections were mounted on glass slides with ProLong Gold Antifade Mountant. The primary antibodies used for immunohistochemistry were as follows: anti-Iba1 (1:500, 019-19741, Fujifilm Wako), anti-MC1 (1:500, a kind gift from P. Davies). Images for MC1 and Iba1 quantification were acquired on Keyence BZ-X700 microscope using 10x objective and analyzed with ImageJ (NIH)[71]. All images were processed with the auto local threshold Phansalkar plugin. Regions of interest including the hippocampus and cortex were hand-traced. MC1+ areas were measured by ImageJ, whereas microglia numbers were counted with the Analyze Particles function. Images for 3D microglia reconstruction were acquired using confocal microscope LSM880 (ZEISS) at 40× magnification. Three fields per mouse of CA3 hippocampal region were taken. 3D structure of microglia was reconstructed using the Imaris software as described before[48]. Experimenters performing imaging and quantification were blinded.

**Endotoxin detection.** Endotoxin levels in tau fibrils, monomers, and LPS were detected using an endotoxin detection kit following the manufacturer's protocol (GenScript ToxinSensor™ Chromogenic LAL Endotoxin Assay Kit). All tau samples had endotoxin levels of <1.0 EU/mL at working concentration.

**Tau fibril stimulation, uptake, and clearance assay.** K18/PL and full-length tau fibrils were synthesized and labeled with Alexa Fluor 647 (K18/PL) as described before[50]. All in vitro high-content assays were performed in 96-well PDL plates (655946, Greiner) with primary microglia at a density of 25,000 cells/well ($Ikbkb^{+/+}$ and $Ikbkb^{-/-}$) or 20,000 cells/well ($Ikbkb^{WT}$ and $Ikbkb^{CA}$). For NF-κB reporter stimulating assay, wild-type microglia were infected with Lenti-κB-dEGFP virus for 24 h and then incubated with tau fibrils, monomers, and LPS at desired concentrations for an additional 24 h. Stimulated microglia were then fixed for high-content analysis. For RNA-seq analysis, wild-type microglia were plated in PDL plates for 24 h before being stimulated with 2 ug/ml full-length tau fibrils for another 24 h. RNA of microglia was then isolated for RNA-seq analysis. For tau fibril uptake assay, microglia were plated for 24 h and then incubated with Alexa-647 labeled K18/PL fibrils (2.5ug/ml) for 6 or 24 h. Media were changed to DMEM containing 0.01% trypsin and incubated for 5 min to remove tau fibrils stuck on the surface of microglia[72]. Accumulated fluorescent tau fibrils were quantified by high-content assay. Lysosomes were labeled with Dextran-FITC (10,000 MW, D1820, ThermoFisher) as described before[42]. Briefly, microglia were incubated with 3 mg/ml Dextran-FITC for 16 h and then chased for at least 3 h with fresh media before being fed with fluorescence tau fibrils. After immunostaining with Iba1, representative images were taken by confocal microscope LSM880 (ZEISS) at 60× magnification. For the clearance assay, microglia that took up tau fibrils for 24 h were cleaned by DMEM/0.01% trypsin, followed by 10% FBS/DMEM incubation for an additional 6 h and 24 h before harvesting for high-content assay to quantify remaining fluorescent tau fibrils.

**High-content analysis assay.** High-content analysis assay, including immunostaining, image acquiring, and automated analysis, was used to unbiasedly detect and quantify immunofluorescence signals in cultured cells[73]. After treatment and trypsin cleaning, microglia were fixed with a conditioned medium containing 4% paraformaldehyde, permeabilized with 0.1% Triton X-100, and incubated for 1 h in a blocking solution containing DPBS, 0.01% Triton X-100, and 5% NGS. The cells

were then incubated in a blocking solution containing anti-Iba1 antibody overnight at 4 °C, followed by incubation with a secondary antibody for 1 h. Nuclei were labeled with Hoechst (H3570, Invitrogen). Images from individual wells were acquired with a fully automated ArraySan XTi high-content system (Thermo Fisher) and a 20× objective. Twenty-five fields containing 3000–5000 microglia and covering the total well area were captured. Images were auto-quantified with CellHealthProfile module of Cellomics software (Thermo Fisher). Iba1 signal was used to define the quantification area and count microglia number. Total intensity of dEGFP or fluorescence tau fibrils within the Iba1 area were quantified and normalized to microglia number in each well.

**Isolation of the sarkosyl-insoluble fraction from the human brain (AD-tau).** Frontal cortical tissue from AD individuals was obtained from the New York Brain bank of Columbia University and are not considered identified "human subjects" and not subjected to IRB oversight. All brains were donated after consent from the next of kin or an individual with legal authority to grant such permission. The institutional review board of Weill Cornell Medicine has determined that clinicopathologic studies on deidentified postmortem tissue samples are exempt from human subject research according to Exemption 45 CFR 46.104(d)(2). The sarkosyl-insoluble fraction of AD cortical tissue was prepared using the method described by Greenberg and Davies[74] with minor modification[44]. Briefly, brain tissue was homogenized with cold homogenization buffer (10 mM Tris/1 mM EGTA/0.8 M NaCl/10% sucrose, pH7.4) in a Teflon glass homogenizer followed with centrifugation at 27,200 × g for 20 min at 4 °C. The supernatant was subjected to extraction with 1% (wt/vol) N-lauroylsarcosine in the presence of 1% (vol/vol) 2-mercaptoethanol at 37 °C for 2 h. After centrifugation at 108,000 × g for 30 min at room temperature, the pellet was rinsed three times with PBS and then dissolved in PBS and stored at −80 °C. This preparation is called the sarkosyl-insoluble AD-tau.

**Microglial AD-tau uptake assay.** Microglia were seeded on the eight-well-chamber slides ($1 \times 10^5$ cells/well) and incubated with 1 μg/ml AD-tau for 2 h in DMEM/10% FBS followed by 0.01%Trypsin clearance, PBS wash, and 4% PFA fixation. After immunostaining with AT8 (1:500) and Alexa Fluor 568 conjugated goat IgG against mouse (Invitrogen# A-11004, 1:200), the cells were stained with Wheat Germ Agglutinin (WGA) conjugated with Alexa Fluor™ 594 (red) (1 μg/ml), and nuclei by DAPI (blue). Representative images were taken by confocal microscope LSM880 (ZEISS) at 60× magnification.

**Microglial AD-tau release assay.** Microglia were seeded on 24-well plate ($3 \times 10^5$ cells per well) in DMEM/10% FBS for 24 h before experiments. Following 2 h pretreatment with DMSO or 0.1μM TPCA-1 (Tocris Bioscience), cells were incubated with 1 μg/ml AD-tau in the presence of DMSO or TPCA-1 for another 2 h. After removing AD-tau-containing medium, 0.01% trypsin was added to cells and incubated for 3 min to remove extracellular tau followed by DMEM/20% FBS to stop digestion. After washing with DMEM/10% FBS twice, cells were incubated with fresh DMEM/10% FBS for 24 h. The conditional medium was harvested. The cells were lysed in RIPA buffer with protease cocktail and halt phosphatase inhibitors and sonicated clear before analysis.

**Tau aggregation assay.** HEK293 expressing human Tau RDP301S FRET biosensor (ATCC CRL-3275) were plated on poly-D-lysine-coated coverslips at a density of $1 \times 10^5$ cells /well in a 24-well cell culture plate in DMEM/10% FBS supplemented with 100μg/ml penicillin and streptomycin. After 24 h, cells were incubated with the conditional medium of microglia treated with or without 2.5 μg/ ml AD-tau for 24 h. After fixation with 4% formaldehyde for 15 min at room temperature, cells were mounted and images were captured under filters for CFP using Zeiss Apotome microscope. Ten images per condition from two biological replicates were taken for quantification using ImageJ. High-resolution image of HEK293 with tau aggregation after 24 h treatment of AD-tau microglia CM was taken by Zeiss LSM880 under 60× len.

**AT8 sandwich ELISA.** High binding 96-well ELISA plates were coated with AT8 (2 μg/ml) (Thermo Fisher Scientific, MN1020) in PBS overnight. The plates were washed with TBST and blocked with StartingBlock buffer (Thermo Fisher Scientific, 37538) for 1 h at RT. The medium and cell lysate were diluted with SuperBlock (Thermo Fisher Scientific, 37516) and loaded on plates for 2 h incubation followed by 1 h incubation with Biotin-HT7 (Thermo Fisher Scientific MN1000B, 1:200). The AT8-positive signals were detected by streptavidin-HRP followed by chromogenic reaction with TMB substrate and reading by Biotek Synergy HT microplate reader at 450 nm.

**Measurement of CMA activity.** Cultures of primary microglia were prepared as described above from KFERQ-mouse model[75] expressing a CMA fluorescent reporter[52]. Microglia were incubated with K18/PL tau fibrils (1μg/ml) in the presence of DMSO or 0.1 μM TPCA-1 for 16 h followed by 20 mM $NH_4Cl$ and 100 mM leupeptin for 8 h to prevent degradation of the reporter translocated into lysosomes during that time. CMA activity was estimated from the degradation of

the reporter as the increase in the intensity of KFERQ-Dendra fluorescence upon blockage of lysosomal proteolysis. The association of KFERQ-Dendra to lysosomes was confirmed by colocalization with the endo/lysosomal marker LAMP1.

**Immunostaining and imaging for CMA activity**. Microglia grown on a coverslip were fixed by 4% PFA, permeabilized with 0.3% Triton X-100, and then blocked with blocking solution (5% normal goat serum, 0.3% Triton X-100) for 1 h at RT. The coverslips were then incubated with primary antibodies in the same blocking solution at 4 °C overnight. The primary antibodies used for immunofluorescence were rabbit anti-dendra2 (1:1000, antibodies, Cat# ABIN361314) or rat anti-lamp1 (1:1000, Hybridoma bank, Cat # 1d4b). After brief washing with PBS (for 5 min, three times), the coverslips were incubated with secondary antibodies for 1 h at RT. The secondary antibodies were Alexa Fluor 488- or Cy5- conjugated goat IgG against rabbit and rat (1:500). Nuclei were co-stained with Hoechst. After brief washing (for 5 min, three times), the coverslips were mounted on slide glasses with ProLong Diamond antifade reagent (Thermo Fisher Scientific, P36965). Confocal images of sixty cells from three independent experiments were acquired with a laser scanning microscope TCS SP8 (Leica), and processed with LAS X software (Leica). ImageJ Software (v.2.1.0) (NIH) was used for image processing and analysis. DiAna plugin in ImageJ was used for colocalization analysis.

**High-throughput bulk RNA-sequencing**. The following sample sets were prepared for bulk RNA-seq analysis. Set 1 (related to Figs. 1a, b, g, i, 2c–e, supplementary Fig. 1c): $N = 4$ replicates of wild-type mouse primary microglia treated with vehicle or 2 μg/ml full-length tau fibrils for 24 h; Set 2 (related to Fig. 1e, f, h, j, supplementary Fig. 1d): N = 4 replicates of adult microglia isolated from 11-month-old male non-transgenic control mice and PS19 mice; Set 3 (related to Fig. 2c–e, h–j, supplementary Fig. 2): $N = 5$ replicates of primary microglia with endogenous IKKβ expression or IKKβCA overexpression and $N = 4$ replicates of primary microglia with wild-type IKKβ expression or deletion of IKKβ; Set 4 (related to Fig. 6, Supplementary Fig. 5): $N = 4$ replicates of $Ikbkb^{+/+}$, $Ikbkb^{-/-}$, $Ikbkb^{WT}$, $Ikbkb^{CA}$, $Ikbkb^{+/+}$; $P301S+$, and $Ikbkb^{-/-}$; $P301S+$ male mice. For all sample sets, total RNA was extracted using Quick-RNA miniprep (R1055, Zymo Research). RNA quality was examined by a 2100 Bioanalyzer Instrument (Agilent Genomics). RNA samples with RNA integrity numbers greater than 8 were used for complementary DNA library construction. For sample set 1–3, cDNA library generation was done using the QuantSeq 3′ mRNA-Seq Library Prep Kit (FWD) for Illumina (K01596, Lexogen) following the manufacturer's instructions. The qualities of the cDNA library were assessed using Qubit 2.0 fluorometer to determine the concentrations and 2100 Bioanalyzer Instrument to determine insert size. cDNA library samples were then sequenced with the HiSeq 4000 System (Illumina). For sample set 4, cDNA library generation and RNA-seq service were performed by Novogene (Novogene Co., Ltd, Sacramento, California). Briefly, oligo(dT) beads were used to enrich mRNA. After chemical fragmentation, the cDNA libraries were generated using NEBNext Ultra RNA Library Prep Kit for Illumina (E7530S, New England Biolabs). The quality of the cDNA library was also assessed by Qubit assay for concentration, Bioanalyzer 2100 for insert size, and qPCR for effective library size. cDNA library samples were then sequenced with the HiSeq 4000 at PE150. On average, 0.02–0.03% error rate was detected. Raw reads were filtered by (1) removing reads containing adaptors; (2) removing reads containing $N > 10\%$ (N represents base that could not be determined); (3) The Qscore (Quality value) of over 50% bases of the read is ≤5.

**Data analyses of bulk RNA-seq**. For sample sets 1–3, RNA-seq reads were mapped using the BlueBee genomics platform and the GENCODE GRCm38 mouse genome (Lexogen QuantSeq 2.2.3) as a reference genome. For sample set 4, RNA-seq read mapping was performed using the STAR program[76] and the same GENCODE GRCm38 mouse genome was used as reference. The total mapping rate is more than 95% and the unique mapping rate is >90%. The read count table was generated with the RSEM program[77]. Differential gene expression was calculated with R package edgeR[78] and limma[79]. Counts were normalized with the trimmed mean normalization method[80]. FDR was calculated using the Benjamini–Hochberg method. DE genes were defined with FDR less than 0.05 in comparison to the control group.

**Isolation of nuclei from frozen mouse brain tissue**. The protocol for isolating nuclei from frozen postmortem brain tissue was adapted from a previous study with modifications[54,81]. All procedures were done on ice or at 4 °C. In brief, mouse brain tissue was placed in 1500 μl of Sigma nuclei PURE lysis buffer (Sigma, NUC201-1KT) and homogenized with a Dounce tissue grinder (Sigma, D8938-1SET) with 15 strokes with pestle A and 15 strokes with pestle B. The homogenized tissue was filtered through a 35 μm cell strainer and was centrifuged at $600 \times g$ for 5 min at 4 °C and washed three times with 1 ml of PBS containing 1% BSA, 20 mM DTT, and 0.2 U μl$^{-1}$ recombinant RNase inhibitor. Then the nuclei were centrifuged at $600 \times g$ for 5 min at 4 °C and resuspended in 500 μl of PBS containing 0.04% BSA and 1× DAPI, followed by FACS sorting to remove cell debris (Supplementary Fig. 7h). The FACS-sorted suspension of DAPI-stained nuclei was counted and diluted to a concentration of 1000 nuclei per microliter in PBS containing 0.04% BSA.

**Droplet-based single-nuclei RNA-seq**. For droplet-based snRNA-seq, libraries were prepared with Chromium Single Cell 3′ Reagent Kits v3 (10× Genomics, PN-1000075) according to the manufacturer's protocol. cDNA and library fragment analysis was performed using the Agilent Fragment Analyzer systems. The snRNA-seq libraries were sequenced on the NovaSeq 6000 sequencer (Illumina) with 100 cycles.

**Analysis of droplet-based single-nuclei RNA-seq**. Gene counts were obtained by aligning reads to the mouse genome (mm10) with Cell Ranger software (v.3.1.0) (10× Genomics). To account for unspliced nuclear transcripts, reads mapping to pre-mRNA were counted. Cell Ranger 3.1.0 default parameters were used to call cell barcodes. We further removed genes expressed in no more than two cells, cells with a unique gene count over 8000 or less than 300, and cells with a high fraction of mitochondrial reads (>5%). Potential doublet cells were predicted and removed using DoubletFinder[55] for each sample. Normalization and clustering were done with the Seurat package v3.0.1[82]. In brief, counts for all nuclei were scaled by the total library size multiplied by a scale factor (10,000), and transformed to log space. A set of 2000 highly variable genes was identified with FindVariableFeatures function based on a variance stabilizing transformation (vst). Principal component analysis (PCA) was done on all genes, and t-SNE was run on the top 15 PCs. Cell clusters were identified with the Seurat functions FindNeighbors (using the top 15 PCs) and FindClusters (resolution = 0.1). For each cluster, we assigned a cell-type label using statistical enrichment for sets of marker genes and manual evaluation of gene expression for small sets of known marker genes. The subset() function from Seurat was used to subset microglia-only cells. Differential gene expression analysis was done using the FindMarkers function and MAST[83].

**Cell trajectory using Monocle3**. For microglial trajectory analysis, the microglial population was first isolated from the other cell types using the previously identified cell types. A separate Seurat object was created for microglia, followed by normalization with a scale factor of 10,000. FindVariableFeatures function was run again to identify the most variable gene-specific for microglia. The microglial Seurat object was then converted into a Monocle3 object with as.cell_data_set function. Size factor estimation of the new CDS (cell dataset) was performed using estimate_size_factors function with default parameters. Further processing of the CDS was carried out using preprocess_cds function with the num_dim parameter set to 9. UMAP was then performed to reduce the dimensionality of the data. Cell clusters were then visualized with cluster_cells function with the parameter K equals to 9. learn_graph function was used to determine the trajectory and the $Ikbkb^{-/-}$; $P301S+$ cluster was selected as the origin of the trajectory.

**Gene network and functional enrichment analysis**. Gene network and functional enrichment analysis were performed by QIAGEN's Ingenuity® Pathway Analysis (IPA®, QIAGEN Redwood City, www.qiagen.com/ingenuity) or by GSEA with molecular signatures database (MSigDB)[84,85]. Significant DEGs and their log2 fold change expression values and FDR were inputted into IPA for identifying canonical pathways, biological functions, and upstream regulators. Upregulated or down-regulated significant DEGs were inputted into GSEA (http://www.gsea-msigdb.org/gsea/msigdb/annotate.jsp) to identify hallmark and gene ontology pathways. The p-value, calculated with the Fischer's exact test with a statistical threshold of 0.05, reflects the likelihood that the association between a set of genes in the dataset and a related biological function is significant. A positive or negative regulation z-score value indicates that a function is predicted to be activated or inhibited. Transcription factors (Supplementary Fig. 5 and Supplementary Fig. 9) were predicted using TRRUST reference database of human transcriptional regulatory interactions[86].

**Plexxikon drug administration**. Plexxikon diet containing 1200 mg/kg PLX5622 compound (Plexxikon Inc., Berkeley) was used as the sole food source for 2 weeks to deplete microglia before inoculation of tau seeds. Control diet with the same base formula but no compound was used for the control group. Mice were fed with plexxikon diet throughout the whole experiment to suppress microglial repopulation[48].

**Preparation of PSP brain extract**. Human brain tissues from PSP cases, which were diagnosed based on accepted neuropathology criteria, were obtained from UCSF Memory and Aging Center. All brains were donated after consent from the next of kin or an individual with legal authority to grant such permission. The use of postmortem brain tissues for research was approved by the University of California, San Francisco's Institutional Review Board with informed consent from patients or their families. The institutional review board has determined that clinicopathologic studies on deidentified postmortem tissue samples are exempt from human subject research according to Exemption 45 CFR 46.104(d)(2). Brain extracts were prepared as previously described[87]. Briefly, PSP brain tissue was homogenized at 10% (w/v) in sterile phosphate-buffered saline (PBS) with protease, phosphatase, and HDAC inhibitors. After brief sonication (power 40, 5 min, 10 s cycles), the homogenates were centrifuged at $3000 \times g$ for 5 min at 4 °C. The protein concentration of the supernatant was measured by BCA assay and then aliquoted and frozen at −80 °C until use.

**Stereotaxic injection**. The stereotaxic surgery was performed as previously described[88]. Briefly, 3-month-old mice that had been fed with PLX or a control diet or injected with tamoxifen were anesthetized with 2% isoflurane by inhalation during the surgery. When deeply anesthetized, mice were secured on a stereotaxic frame (Kopf Instruments) and were unilaterally injected into the right hippocampus using CA1 coordinates (bregma, −2.5 mm; lateral, +2 mm; and depth, −1.8 mm) with a Hamilton syringe under aseptic conditions. The following tau seeds were inoculated: (1) 3 μL of 4.3 μg/ul PSP brain extracts for microglia-depleted PS19 mice (Supplementary Fig. 4); (2) 2 μl of 2.5 μg/ul K18/PL tau fibrils for microglia-depleted PS19 mice (Fig. 4a–c) and PS19 mice with microglial NF-κB inactivation (Fig. 4f–h); (3) 2 μL of 0.2 μg/ul K18/PL tau fibrils for PS19 mice with microglial NF-κB activation (Fig. 4k–m). Same volumes of PBS were injected into PS19 mice or the same amount of tau seeds were injected into non-transgenic mice as control. Mice were monitored during the anesthesia until recovery. After 3 months (for PSP brain extracts) or 1 month (for K18/PL tau fibrils) of spreading, mice were perfused for whole-brain immunohistochemistry of AT8 or MC1 tau as described.

**Morris water maze**. Morris water maze studies were conducted during daylight hours. The water maze consisted of a pool (122 cm in diameter) containing opaque water (20 ± 1 °C) and a platform (14 cm in diameter) submerged 1.5 cm below the surface. Hidden platform training (days 1–6) consisted of 12 sessions (two per day, 2 h apart), each with two trials. The mouse was placed into the pool at alternating drop locations for each trial. A trial ended when the mouse located the platform and remained motionless on the platform for 5 s, for a maximum of 60 s per trial. Mice that failed to find the platform within the 60 s trial were led to it and placed on it for 15 s. Probe trails were conducted 72 h after the final hidden training. Mice were returned to the pool with a drop location that was 180° opposite of the original target platform location in the absence of the hidden platform. Performance was measured with EthoVision video-tracking (Noldus Information Technology). Visible platform training, where the platform was cued with a mounted black-and-white striped mast, was conducted for three sessions after the conclusion of probe trials. Pre-set criteria for exclusion from the analysis included floating and thigmotaxic behaviors, neither of which was observed in current studies.

**Statistics**. The sample size for each experiment was determined on the basis of previous publications. All in vitro experiments were performed with a minimum of three biological replicates. Mean values from at least three independent experiments were used for computing statistical differences. All in vivo experiments were performed with a minimum of four mice per genotype. All in vivo data were averaged to either individual mouse (microglia number counts), individual section (MC1, AT8 tau), or individual microglia (Imaris morphology analysis), and mean values were used for computing statistical differences. Data visualization was done with Graphpad and R package ggplot2[89]. Statistical analyses were performed with Graphpad prism 9.0 (t-test, one-way and two-way ANOVA) (Graphpad, San Diego, California), STATA 12 (multilevel mixed-effects model) (StataCorp), R (R Foundation for Statistical Computing, Vienna, Austria). Values are reported as mean ± standard error of the mean (SEM) or standard deviation (SD). The Shapiro-Wilk test of normality and F test to compare variances were applied to datasets when applicable. Unpaired two-tailed or one-tailed t-test was used to compare two groups. Mann–Whitney test was used when the normality test is not passed. One-way ANOVA was used to compare data with more than two groups. Two-way ANOVA was used for groups with different genotypes and/or time as factors. Tukey's and Sidak's post-test multiple comparisons were used to compare the statistical difference between designated groups. The multilevel mixed-effects linear regression model was used when parameters vary at more than one hierarchical level. All P-values of enrichment analysis are calculated by right-tailed Fisher's exact test. $P < 0.05$ was considered statistically significant.

**Reporting summary**. Further information on research design is available in the Nature Research Reporting Summary linked to this article.

## Data availability

All data supporting this study are involved in the paper and its supplementary files. All RNA-seq data were deposited in the Gene Expression Omnibus (GEO) under the following series accession numbers: bulk-tissue RNA-seq GSE198013; mouse single-nuclei RNA-seq, GSE198014. Source data are provided with this paper.

## Code availability

All custom codes used for snRNA-seq data analysis have been archived at Zenodo (https://doi.org/10.5281/zenodo.6336233) and are directly available at https://github.com/lifan36/Wang_et_al_NatC_2021.

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

## Acknowledgements

We thank Dr. Peter Davies from Albert Einstein College of Medicine for MC1 antibody; Dr. Michael Karin at University of California-San Diego and Dr. Katerina Akassoglou at Gladstone Institutes for the *Ikbkb*$^{F/F}$ mice; Dr. Michael Gill at the Gladstone Behavior Core for the MWM test assistance; Dr. Meredith Calvert at the Gladstone Histology and Light Microscopy Core for imaging assistance; Dr. Anke Meyer-Franke at the Gladstone Assay Development and Drug Discovery Core for the high-content assay assistance; Dr. Santiago Sole Domenech and Dr. Fred Maxfield at Weill Cornell Medicine for lysosome tracking; the University of California, San Francisco's Center for Advanced Technology for conducting bulk RNA-sequencing; Jason McCormick and Tomas Baumgartner from Weill Cornell Medicine Flow Cytometry Core Facility for FACS assistance; Dong Xu, Xing Wang, and Adrian Tan from Weill Cornell Genomics Resources Core Facility for performing single-nuclei RNA-sequencing; Rose Horowitz and Jennifer Guo from Weill Cornell for immunohistochemistry. This work was supported in part by the National Institute of Health Grants R01AG051390, U54NS100717, R01AG054214 R01AG072758 (to L.G.), U54NS100717 and Rainwater Foundation (to L.G. and A.M.C.), JPB Foundation (to L.G. and A.M.C.) the National Institute of Aging Grant F30 AG062043-02 and National Institute of Health Grant T32GM007618 (to L.K.), the National Institute of Aging Grant R01 AG064239 (to W.L.), R.R.K. is supported by IRACDA program grant K12 GM102779.

## Author contributions

C.W. and L.G. designed research; C.W., L.F., R.K., L.Z., L.K., M.C., Y.L., D.L., Y.Z., C.C., and W.L. performed experiments; S.M., J.G., L.T.G., W.W.S., and B.M. provided key reagents; C.W., L.F., R.K., L.Z., L.K., M.C., B.L., Y.L., D.L., C.C., A.M.C., W.L., and L.G. analyzed data; C.W., L.F., B.L., W.L., and L.G. wrote the manuscript.

## Competing interests

L.G. is a founder of Aeton Therapeutics, Inc. The remaining authors declare no competing interests.
