## [Peer Review File · Nature Communications]

Reviewers' Comments:

Reviewer #1:

Remarks to the Author:

In this manuscript, Chao Wang et al. explore the effects of manipulating NFkB signaling specifically in microglia for the initiation, spreading and overall neurophysiological and behavioral consequences of tau pathology. The manuscript combines multiple approaches to either enhance or block NFkB signaling, as well as in vitro and in vivo models. The data is well presented, and the methodology is sound, but I believe there are some relevant issues and a few points that could be improved or addressed in more detail.

- My major concern arises from the a priori contradictory data coming from the "acute" model injecting seeds, and the "chronic" PS19 system with no seeds. I find this hard to reconcile as the authors use contradictory arguments along the paper, and do not provide a clear message. First, they do use MC1 pathology as a measure of positive and negative effects upon manipulation of NFkB and that is actually how they decide to further explore the blockade of NFkB in PS19 in a chronic set up. However, they completely disregard MC1 changes in PS19 mice (without seeds) and they even propose that tau inclusions are not toxic ("this indicates that neuronal tau inclusions might not be toxic per se if maladaptive microglial responses are blocked by inactivation of NFkB"). I find this contradictory. If that were the case, why do the authors still rely on quantification of tau inclusions anyway? Why not proposing alternative readouts?
- In figure 3 – phagocytosis analysis – the control groups are very variable across experiments. It actually seems that the mutants remain constant and it is the controls that are either showing a reduction or increment in the phagocytic/clearance capacity. How consistent are these results over independent replicates?
- A number of studies already addressed the impact of manipulating microglia by either depletion () or reduction of proliferation (Mancuso et al., Brain 2019). These studies are complementary to the data presented here and should be discussed accordingly. Additionally, transcriptomic analysis of microglia from tauopathy models has also been performed before (see Friedman et al., Cell Rep). Whereas I understand it is not the primary goal of the manuscript, comparing these newly generated data with that from other tauopathy models will at least inform as to whether the results could be extrapolated to other systems (for example, is NFkB activation a consistent finding in tau models or just present in this particular system).
- In general, I would appreciate if the authors could display all the individual values when they represent the data.
- The authors use a CRE-loxP system to either enhance or repress NFkB (e.g. in figures 2 and 4), however it is unclear what strains are being compared in all the experiments that confirm up or down regulation. Given loxP systems are prone to show some leakiness, it would be helpful to also compare levels to those of wild type microglia.
- It would be useful to see the number of genes that are responsible for the pathway enrichments, as well as the overlap between different pathways. In other words, is the same limited number of genes explaining enrichment of multiple pathways?
- The threshold for significant enrichment for the pathway analysis should be displayed in the figures
- In figure 1, it is not clear how the upstream regulators were defined. Is this based on in silico prediction of transcription factors can potentially explained differences observed at transcriptomic level? What tools and what conditions were used?
- In figure 5, microglial morphology analysis. How many cells from how many mice were analyzed?
- In figure 3, the authors refer to processing but perhaps it would be more accurate to use other terms like protein degradation or digestion? Or proteolysis?

Reviewer #2:

Remarks to the Author:

The study by Wang and colleagues provides novel and impactful findings regarding the role of microglial NFkB activity in response to tau protein stimulation. It is an extensive collection of data that is rigorous and well-controlled. The findings will be of interest to Journal readers. A few minor comments arose during review which could be addressed by additional data or expansion of the

discussion.

It would be interesting to elaborate on the apparent ability of tau to stimulate NFkB-mediated changes independent of fibril state and what differences in uptake or intracellular turnover may exist for each.

Some demonstration of the changes in tau fibrils after microglial uptake and secretion into the media would have been supportive of the idea that microglia process tau into a unique state transmissible between neurons.

Perhaps a demonstration of the biophysical differences or microglial NFkB activation differences between the exogenous tau versus the intraneuronal PS19 brain tau might offer some insight into the contrasting effects on intraneuronal inclusions.

Reviewer #3:

Remarks to the Author:

This manuscript by Chao Wang et al. showed tau fibrils can induce NFkB signaling in microglial cells and that primary microglial cells isolated from PS19 mice also show elevated NFkB pathway activation. The authors further demonstrated constitutive activation of IKBKB (NFkB) in microglial cells reduced intracellular tau fibrils while knocking out Ikbkb increased intracellular accumulation of tau fibrils in vitro. Depletion of microglial cells or knocking out microglial Ikbkb inhibited tau deposition in the cortex and hippocampus in the tau spreading model, whereas Ikbkb overexpression showed the opposite effect in vivo. Ikbkb deficiency further rescued the behavioral deficits and restored the transcriptomic landscape in the PS19 mice, comparable to the wild-type mice. While this is an interesting study, there are several moderate to major weaknesses in terms of cell type consistency (for example, there is no co-localization of MC1 staining with microglia or neuron to support their argument that Ikbkb knockout in PS19 mice reduces neuronal MC1 while increasing microglial MC1 staining) to conclusively establish their interpretation. Finally, there is no direct evidence showing Ikbkb activation upon overexpression (e.g., nuclear p65 staining or any reporter assay – like the one in Fig. 1c) affects tau secretion by neurons and propagation via microglia. These and other concerns are listed below.

Major

1. Fig. 1a-c: It's not clear if the authors used primary human microglia or mouse microglia. Please specify.
2. Fig. 1a-c: Are these synthetic fibrils or purified from human/PS19 brains? Latter may be more relevant than the synthetic version. Also, how specific is this to disease-relevant tau as the authors did not use the pathological version of tau? Does it mean even in the normal brain, these pathways are induced by WT tau? It has been reported that AD-patient-derived tau fibrils exhibited different seeding potency and distinct conformational features from the synthetic, overexpressed tau fibrils.
Guo, J. L., Narasimhan, S., Changolkar, L., He, Z., Stieber, A., Zhang, B., et al. (2016). Unique pathological tau conformers from Alzheimer's brains transmit tau pathology in nontransgenic mice. *J. Exp. Med.* 213, 2635–2654. doi: 10.1084/jem.20160833
3. Were the experiments in Fig. 2c (tau stimulated MG and IkbkbCA MG) done at the same time? Or were those 1200 or so genes from Tau MG group from the previous experiment in Fig. 1a?
4. Not quite sure about the relevance of Ikbkb-CA overexpression system? How is this relevant to tauopathies as the expression is >3 fold more than the basal level?
5. Page 7, line 189, Fig. 3a and b. Ikbkb+/+ and IkbkbWT. What are these mice exactly? Is Ikbkb+/+ equal to Ikbkb^{f/f}? Is Ikbkb+/+ the same as IkbkbWT?
6. Fig. 3e: why is the black line (control group) in 3e is different from the black line (control group) in 3g? Rather than the genotype effect (Ikbkb^{-/-} vs IkbkbCA), changes in controls appear to be making experimental groups show the difference (e.g. 38% vs 62% in controls at 6 h time point). Also, Fig. 3e and one data point in Fig. 3g have no error bars (or if the variance is minimal/close to none?).

7. Supp.Fig.4g: why there is no AT8/MC1 in 4-5 month-old (if they are age-matched as other panels in the figure) PS19 mice (which should have quite a robust basal tau pathology at this age)?
8. Since this entire study hinges on NF- κ B activation, it may be important to show panels of nuclear p-p65 staining in a few representative experiments (specifically for in vivo data sets) to conclusively establish that it is 'activation' of NF κ B - not just overexpression of I κ bb is driving tau spreading.
9. In Fig. 4g-h, there should be more microglial MC1+ tau expected coupled with reduced neuronal MC1+ tau in the ipsilateral cortex of PS19/I κ bb^{-/-} (per data from Fig. 3e-g). That means, the PS19 with I κ bb^{-/-} background doesn't show the same phenotype in accumulating tau in microglia, as in vitro microglia in Fig. 3e-g? One way to confirm this is by doing a double IF for Iba1+MC1 or Iba1+AT8 for panels in Fig. 5c-d and/or Fig.6a-c.
10. Fig. 4f-h shows cortex and k-m shows hippocampus. Can authors include both cortex and hippocampus in both panels for better comparison?
11. There seems to be some disconnect in terms of the consequences of what happens to tau seeds within microglia. In Fig. 3e-g, microglial NF κ B activation (in case of I κ bbCA microglia), leads to reduced intra-microglial tau (i.e. increased tau processing - hopefully, clearance), which is not the case in microglia with inactivated NF κ B (in case of I κ bb^{-/-} MG). If so, then how the results in Fig. 5 on the inactivation of microglial NF κ B showing a rescued phenotype - amidst build-up of tau within microglia per Fig. 3e-g?
12. The behavior tests were done at 8-9 months in Fig.5. While in Fig.6 (9-10 months), I κ bb^{-/-} mice have increased tau inclusion. Are these the same mice used for the behavior tests in Fig. 5? Relevant to comment No. 9 above, microglial/neuronal markers should be used at the same time with MC1 or AT8 staining.
13. Page 13, line 366. "NF- κ B activation could promote microglia to secrete more seeding-competent tau, and thus accelerate the spread of tau pathology." To prove this statement, conditioned media from PS19/ I κ bbCA cells (and controls) should be investigated for 'seeding capable tau' in a seeding assay. As it is stated, there is no data to prove that PS19/ I κ bbCA microglia do secrete 'seedable tau'. Rather current results appear to achieve better clearance of tau in PS19/ I κ bbCA microglia.
14. Microglial NF κ B modulation can also affect synaptic pruning. Have the authors looked into investigating any synaptic markers that would support their behavioral readouts?

Minor:

1. Page 3, line 68: "onset", not "onsite".
2. Fig. 1g: the caption of the figure should be "stimulated", not "stiumuted".

Point-to-point response to the reviewers:

We appreciate the thorough review of our manuscript. Guided by reviewers' comments, we have performed extensive new experiments and addressed essentially all of their concerns as itemed below:

Reviewer #1:

In this manuscript, Chao Wang et al. explore the effects of manipulating NFκB signaling specifically in microglia for the initiation, spreading and overall neurophysiological and behavioral consequences of tau pathology. The manuscript combines multiple approaches to either enhance or block NFκB signaling, as well as in vitro and in vivo models. The data is well presented, and the methodology is sound, but I believe there are some relevant issues and a few points that could be improved or addressed in more detail.

We thank the reviewer for their thorough review that greatly helped us to improve the manuscript.

1. My major concern arises from the a priori contradictory data coming from the “acute” model injecting seeds, and the “chronic” PS19 system with no seeds. I find this hard to reconcile as the authors use contradictory arguments along the paper, and do not provide a clear message. First, they do use MC1 pathology as a measure of positive and negative effects upon manipulation of NFκB and that is actually how they decide to further explore the blockade of NFκB in PS19 in a chronic set up. However, they completely disregard MC1 changes in PS19 mice (without seeds) and they even propose that tau inclusions are not toxic (“this indicates that neuronal tau inclusions might not be toxic per se if maladaptive microglial responses are blocked by inactivation of NFκB”). I find this contradictory. If that were the case, why do the authors still rely on quantification of tau inclusions anyway? Why not proposing alternative readouts?

Responses: The reviewer is completely correct in pointing out the apparent contradictory roles of microglial NF-κB in the acute (seeding) vs. the chronic (non-seeded) PS19 mice. The reviewer's comments prompted us to further clarify these two models. We now included a hypothetical model to summarize our observations as new Figure 8 in the revised manuscript.

In the acute model in young neurons (Fig. 8A), the spread of tau inclusions is driven by the inoculation of exogenous tau seeds. We showed that NF-κB activation accelerates the spread of tau inclusions in vivo and inhibition reduces the release of tau seeds in cultured primary microglia (Revised Fig. 3n). In contrast, in the non-seeded PS19 model of aged neurons (Fig. 8B), age-dependent tau aggregation is determined by tau proteostasis of neurons expressing the transgene. In this model, NF-κB activation in microglia drives functional toxicity while reduces intraneuronal tau aggregates, which could result from enhanced tau exocytosis (or degradation) in these neurons.

New Fig 8: Hypothetical model illustrating microglial NF-κB activity drive tau seeding and spreading in young neurons with exogenous seeds (A) and tau-induced neuronal toxicity and cognitive deficits in aged neurons (B).

We agree completely that tau pathology is multifaceted. While intraneuronal tau load can be pathological, it does not represent all pathological tau species, such as those secreted that can exert toxicity as well. Thus, the amount of intraneuronal tau inclusions may not directly correlate with the extent of neuronal injury. Thus, besides quantifying tau inclusions, we also measured the spatial learning and memory, and transcriptomic alterations induced by altering microglial NF-κB activation status.

2. In figure 3 – phagocytosis analysis – the control groups are very variable across experiments. It actually seems that the mutants remain constant and it is the controls that are either showing a reduction or increment in the phagocytic/clearance capacity. How consistent are these results over independent replicates?

Responses: We thank the reviewer point out the difference of tau clearance capacity between the two control groups in Figure 3. This is due to the lower seeding density (20,000 cells/well) of the control for *Ikbkb^{CA}* (*Ikbkb^{WT}*) compared to the control for *Ikbkb^{-/-}* (*Ikbkb^{+/+}*, 25,000 cells/well). So even though both carry wildtype *Ikbkb* alleles, the tau degradation rates can differ.

For each phagocytosis assay, we performed three to four independent experiments with 4-8 biological replicates and obtained consistent results. Since *Ikbkb^{-/-}* or *Ikbkb^{CA}* microglia were compared to their own littermate wildtype controls at the same seeding density, our conclusion is still sound. We included the seeding density information in the legends of new Fig. 3 and in the Method section.

3. A number of studies already addressed the impact of manipulating microglia by either depletion () or reduction of proliferation (Mancuso et al., Brain 2019). These studies are complementary to the data presented here and should be discussed accordingly. Additionally, transcriptomic analysis of microglia from tauopathy models has also been performed before (see Friedman et al., Cell Rep)....(for example, is NFκB activation a consistent finding in tau models or just present in this particular system).

Response: we thank the reviewer and completely agree. In the revised manuscript, we added the citations of Mancuso et al., Brain 2019 in the revised introduction.

In addition, per reviewer's suggestion, we performed transcriptomic analysis of microglia dataset (GSE93180) from the tau-P301S model published in Friedman et al Cell Rep 2018 (Ref #58) as Supplementary Fig. 8 in the revised manuscript. We also included Wang, H. et al. Mol Neurodegener, 2018, in which the transcriptomic analyses were performed in rTg4510 tau transgenic mouse model. Both studies revealed a significant activation of NF-κB signaling, indicating that this is a common finding across different studies and different tauopathy models

Supplementary Fig. 8 Activation of NF-κB in GSE93180: Hippocampal CD11b cells in Tau-P301S model. DEGs from GSE93180 (<https://www.ncbi.nlm.nih.gov/geo/query/acc.cgi?acc=GSE93180>) were

analyzed by TRRUST transcription factor database (a) and GSEA hallmark pathways (b). NF-κB signaling is predicted to be activated in microglia from Tau-P301S model.

4. In general, I would appreciate if the authors could display all the individual values when they represent the data.

Response: All individual values are now presented in figures.

5. The authors use a CRE-loxP system to either enhance or repress NFκB (e.g. in figures 2 and 4), however it is unclear what strains are being compared in all the experiments that confirm up or down regulation. Given loxP systems are prone to show some leakiness, it would be helpful to also compare levels to those of wild type microglia.

Response: We use a well validated Cx3cr1-CRE-Loxp system to restrict the CRE expression only in microglia in CNS. The specificity of this line was recently comprehensively revalidated by Zhao et.al (<https://doi.org/10.1523/ENEURO.0114-19.2019>), which found no leakiness. Enhanced or deletion of *Ikkkb* gene were confirmed (Fig. 2b,g and Fig. 4e,j) when compared to their own littermate controls that have same loxP sites but no Cre gene. We added a table in Method section to illustrate the correspondence of genotypes and symbols.

Gene symbols	Genotype	IKKβ levels in microglia
Ikkkb ^{+/+}	Cx3cr1 ^{+/+} ; Ikkkb ^{F/F}	Normal
Ikkkb ^{-/-}	Cx3cr1 ^{CreERT2/+} ; Ikkkb ^{F/F}	Deleted
Ikkkb ^{WT}	Cx3cr1 ^{+/+} ; IkkkbCA ^{F/F}	Normal
Ikkkb ^{CA}	Cx3cr1 ^{CreERT2/+} ; IkkkbCA ^{F/F}	High

6. It would be useful to see the number of genes that are responsible for the pathway enrichments, as well as the overlap between different pathways. In other words, is the same limited number of genes explaining enrichment of multiple pathways?

Response: We thank the reviewer's suggestion. For each pathway enrichment analysis, we now include a supplementary table to specify: 1) number of genes for each pathway; 2) gene symbol; 3) the occurrence of each gene in each pathway. These tables include: Supplementary table 3 for Fig. 1b, Supplementary table 5 for Fig. 1f, Supplementary table 7 for Fig. 2e, Supplementary table 9 for supplementary Fig. 2a-c, Supplementary table 12 for Fig. 6f. Attached is an example of supplementary table 5 for Fig. 1f. As shown in this table, different genes are responsible for different pathways.

7. The threshold for significant enrichment for the pathway analysis should be displayed in the figures.

Responses: The threshold for significant cutoff/enrichment of the transcriptomic analyses in this study is set as 0.05 and displayed as dashed line in all figures now (-log(p-value)/FDR≥1.3). We specify this in the figure legends of Fig. 1b.

8. In figure 1, it is not clear how the upstream regulators were defined. Is this based on in silico prediction of transcription factors can potentially explained differences observed at transcriptomic level? What tools and what conditions were used?

Responses: The upstream regulator analysis was performed by QIAGEN's Ingenuity Pathway Analysis software, which identifies the cascade of upstream transcriptional regulators that can explain the observed gene expression changes in a dataset. The prediction is based on prior knowledge of expected effects between transcriptional regulators and their target genes stored in the Ingenuity Knowledge Base. Known targets of transcription regulators and their direction of changes in our dataset were analyzed to predict likely relevant transcription regulator. We further clarify this point in the revised Result section, and added a reference for the causal analysis approaches in Ingenuity Pathway Analysis in our revision (Ref #36).

9. In figure 5, microglial morphology analysis. How many cells from how many mice were analyzed?

Responses: The information is now included in the legend of Fig. 5 as 'N=4 mice per genotype (4 sections per mouse) were imaged and a total of 893 (*Ikbkb*^{+/+}), 902 (*Ikbkb*^{-/-}), 3455 (*Ikbkb*^{+/+};P301S+) and 2143 (*Ikbkb*^{-/-};P301S+) microglia were analyzed'.

10. In figure 3, the authors refer to processing but perhaps it would be more accurate to use other terms like protein degradation or digestion? Or proteolysis?

Responses: We thank the reviewer for the suggestion. Consistent with previous studies (Hopp et al., 2018; Luo et al., 2015; Pérez et al., 2019), in a new pulse-chase experiment, we observed that AD brain-derived tau (AD-tau) can be secreted by microglia with about ~55% of internalized tau processed by microglia (Revised Fig. 3k). Moreover, we showed that inactivation of NF-κB reduced the amount of extracellular tau (Fig. 3n). Thus, our findings support the likely involvement of NFκB activity in microglial exocytosis/secretion besides degradation. Therefore, we believe "processing" is a more accurate term to describe a combination of tau uptake, degradation/proteolysis, and exocytosis/secretion.

Responses to Reviewer #2 (Remarks to the Author):

The study by Wang and colleagues provides novel and impactful findings regarding the role of microglial NFκB activity in response to tau protein stimulation. It is an extensive collection of data that is rigorous and well-controlled. The findings will be of interest to Journal readers. A few minor comments arose during review which could be addressed by additional data or expansion of the discussion.

We thank the reviewer for the positive comments.

1. It would be interesting to elaborate on the apparent ability of tau to stimulate NFκB-mediated changes independent of fibril state and what differences in uptake or intracellular turnover may exist for each.

Response: We agree with reviewer that systematic dissection of how microglia respond to different conformations of tau (monomers, oligomers, and fibrils) is important. However, given the complexity of the topic, our current study mainly focuses on the interaction of tau fibrils with microglia.

2. Some demonstration of the changes in tau fibrils after microglial uptake and secretion into the media would have been supportive of the idea that microglia process tau into a unique state transmissible between neurons.

Response: We thank the reviewer for the excellent suggestion. We have performed new experiments to assess the ability of microglia-processed tau to induce seeding in a HEK cell-based reporter line (Holmes et al, 2014). As described in the revised Fig. 3, we showed that sakosyl-insoluble tau isolated from AD brains, can be taken up, processed, degraded and released into the media by microglia, which possess tau seeding activity (Fig. 3h-m). Notably, we showed that inactivation of NFκB reduced tau release from microglia (Fig. 3n).

Revised Fig. 3.

(h) Schematic diagram illustrating the isolation procedures of human AD-tau.

(i) Schematic diagram illustrating collection of conditional medium and lysates from microglia pulsed with AD-tau for 2 hrs followed by 24 hrs chase in tau-free medium.

(j) Uptake of AD-tau by microglia after 2 hrs pulse. Intracellular pTau was detected by immunofluorescence using AT8 antibody (green), cell membranes by Wheat Germ Agglutinin (WGA) conjugated with Alexa Fluor™ 594 (red), and nuclei by DAPI (blue). Scale bar: 10 μm

(k) After 2 hrs pulse with AD-tau followed by 24 hrs chase, the secreted pTau in the conditional medium and the intracellular pTau were determined by AT8 ELISA and shown as the percentage of total pTau loaded in microglia after 2 hrs pulse. N=4 wells. Values are mean ± SEM.

(l) Representative images of tau aggregation induced by conditional medium from microglia treated with AD-tau in HEK biosensor cells. Insert: High magnification confocal image of HEK biosensor cells with tau aggregation. Nuclei was detected by DAPI. Scale bar: 50 μm, insert: 15 μm.

(m) Quantification of tau aggregation positive cells. A representative dataset from 3 independent experiments. N=10 fields from two wells, values are mean ± SEM, **p<0.01, student t-test with unequal variance.

(n,o) NF-κB inhibitor TPCA-1 treated microglia were pulsed 2 hrs with AD-tau followed by 24 hrs chase. The pTau released to medium **(n)** and intracellular pTau **(o)** were determined by AT8 ELISA. A representative dataset from 2 independent experiments. N=4 wells, values are mean ± SEM, relative to DMSO control. *p<0.05, student t-test.

3. Perhaps a demonstration of the biophysical differences or microglial NFκB activation differences between the exogenous tau versus the intraneuronal PS19 brain tau might offer some insight into the contrasting effects on intraneuronal inclusions.

Response: We thank the reviewer for the helpful suggestion. We agree that there could be biophysical differences between the exogenous tau versus the intraneuronal PS19 brain tau. Besides the possible biophysical difference, another obvious difference is the timeline (acute vs. chronic). We elaborate the differences of these two models in a diagram (new Fig. 8, also see

response to the #1 of Reviewer 1). In the acute setting inoculated with exogenous tau seeds, microglial-mediated tau processing (uptake, processing and secretion) drives the spread of tau inclusions. In the scenario of aged PS19 brain, tau aggregation driven by chronic expression of the transgene, instead of spreading from one brain subregion. These additional key differences could also contribute to the differential effects of microglial NF- κ B on neuronal tau proteostasis.

Reviewer #3 (Remarks to the Author):

This manuscript by Chao Wang et al. showed tau fibrils can induce NF κ B signaling in microglial cells and that primary microglial cells isolated from PS19 mice also show elevated NF κ B pathway activation. The authors further demonstrated constitutive activation of I κ BK β (NF κ B) in microglial cells reduced intracellular tau fibrils while knocking out I κ bkb increased intracellular accumulation of tau fibrils in vitro. Depletion of microglial cells or knocking out microglial I κ bkb inhibited tau deposition in the cortex and hippocampus in the tau spreading model, whereas I κ bkb overexpression showed the opposite effect in vivo. I κ bkb deficiency further rescued the behavioral deficits and restored the transcriptomic landscape in the PS19 mice, comparable to the wild-type mice.

1. While this is an interesting study, there are several moderate to major weaknesses in terms of cell type consistency (for example, there is no co-localization of MC1 staining with microglia or neuron to support their argument that I κ bkb knockout in PS19 mice reduces neuronal MC1 while increasing microglial MC1 staining) to conclusively establish their interpretation.

Response: The reviewer raised a good point. Interestingly, in human tauopathies, besides neurons, tau inclusion has been detected in astrocytes, but so far no solid or consistent evidences have been presented for microglia (Kahlson and Colodner, 2015). Consistent with these observations in human brains, there has been no reports of MC1 positive microglia in PS19 brains. One likely explanation is that as professional phagocytes, microglia mediate efficient proteolysis and secretion of tau, precluding detection of intracellular tau inclusions in microglia.

2. Finally, there is no direct evidence showing I κ bkb activation upon overexpression (e.g., nuclear p65 staining or any reporter assay – like the one in Fig. 1c) affects tau secretion by neurons and propagation via microglia. These and other concerns are listed below.

Response: We appreciate the reviewer's constructive comment and performed new experiments to address this concern. In this revision, we showed that NF- κ B inactivation reduced tau secretion from microglia (Revised Fig. 3h-o, also see response #2 to reviewer 2), suggesting that microglial NF- κ B activation could contribute to tau propagation via modulating microglial responses. We agree that it will be insightful to dissect the molecular mechanisms underlying microglial NF- κ B activation on neuronal tau secretion. This may involve a more complex in vitro setting by coculturing neurons and microglia and more sensitive assays and will be further investigated in our future study.

3. Fig. 1a-c: It's not clear if the authors used primary human microglia or mouse microglia. Please specify.

Response: Primary mouse microglia were used for all in vitro experiments and specified in the Result section and figure legends now.

4. Fig. 1a-c: Are these synthetic fibrils or purified from human/PS19 brains? Latter may be more relevant than the synthetic version. Also, how specific is this to disease-relevant tau as the authors did not use the pathological version of tau? Does it mean even in the normal brain, these pathways are induced by WT tau? It has been reported that AD-patient-derived tau fibrils

exhibited different seeding potency and distinct conformational features from the synthetic, overexpressed tau fibrils. (Guo, et al J. Exp. Med. 2016)

Responses: We completely agree with the reviewer. We showed that both recombinant tau fibrils and tau fibrils from human brains, including those from PSP patient brain lysates, can drive microglial-dependent tau seeding and spreading (supplementary Fig.4e-g).

In the revision, we isolated disease-relevant AD-tau (Revised Fig. 3h). Using sarkosyl-insoluble tau prep derived from AD brains, our new experiments showed that NF-κB promotes microglial processing and secretion of tau with seeding activity (Revised Fig. 3, also see response #2 to Reviewer 2). We showed a potent seeding activity of AD-tau incubated with microglia using HEK human tauRDP301S biosensor cells (Holmes et al., 2014) (Revised Fig.3l-m). Moreover, NF-κB inhibitor TPCA-1 significantly reduced AT8-positive p-tau released from microglia (Revised Fig. 3n). Thus, inhibition of NF-κB reduces the microglial processing and release of tau species with seeding activity.

5. Were the experiments in Fig. 2c (tau stimulated MG and *Ikbkb*^{CA} MG) done at the same time? Or were those 1200 or so genes from Tau MG group from the previous experiment in Fig. 1a?

Response: The tau stimulated MG group in Fig. 2c is done from the same experiment in Fig. 1a. The stimulations and library preparations of tau stimulated MG and *Ikbkb*^{CA} MG were done at different times using the same method (QuantSeq 3' mRNA-Seq Library Prep Kits). We have further clarified in the Method section.

6. Not quite sure about the relevance of *Ikbkb*-CA overexpression system? How is this relevant to tauopathies as the expression is >3 fold more than the basal level?

Response: We used the *Ikbkb*-CA system to establish direct outcome of activating NF-κB in microglia. The system is relevant for the tauopathy model for the following reasons: 1) Tau fibrils at 2ug/ml increased NF-κB activity > 3 folds compared to untreated cells (Fig. 1c-d). 2) Our transcriptome study of PS19 microglia indicated that NF-κB pathway is activated and *Ikbkb* is predicted as main upstream regulator (Fig. 1e-g). Therefore *Ikbkb*-CA may mimic microglial NF-κB activation in response to brain tau pathology.

7. Page 7, line 189, Fig. 3a and b. *Ikbkb*^{+/+} and *Ikbkb*^{WT}. What are these mice exactly? Is *Ikbkb*^{+/+} equal to *Ikbkb*^{f/f}? Is *Ikbkb*^{+/+} the same as *Ikbkb*^{WT}?

Response: To clarify the correspondence of genotypes and gene symbols used in this manuscript, we added a table in the Method section. Specifically, both *Ikbkb*^{+/+} and *Ikbkb*^{WT} are wildtype controls. We used +/+ and WT to indicate that they're littermates controls for *Ikbkb*^{-/-} and *Ikbkb*^{CA}

Gene symbols	Genotype	IKKβ levels in microglia
Ikbkb ^{+/+}	Cx3cr1 ^{+/+} ; Ikbkb ^{F/F}	Normal
Ikbkb ^{-/-}	Cx3cr1 ^{CreERT2/+} ; Ikbkb ^{F/F}	Deleted
Ikbkb ^{+/+} ; P301S ⁺	P301S ⁺ ; Cx3cr1 ^{+/+} ; Ikbkb ^{F/F}	Normal
Ikbkb ^{-/-} ; P301S ⁺	P301S ⁺ ; Cx3cr1 ^{CreERT2/+} ; Ikbkb ^{F/F}	Deleted
Ikbkb ^{WT}	Cx3cr1 ^{+/+} ; Ikbkb ^{CA} ^{F/F}	Normal
Ikbkb ^{CA}	Cx3cr1 ^{CreERT2/+} ; Ikbkb ^{CA} ^{F/F}	High

Ikkkb^{WT};P30IS+	P30IS+;Cx3cr1^{+/+};IkkkbCA^{F/F}	Normal
Ikkkb^{CA}; P30IS+	P30IS+;Cx3cr1^{CreERT2/+};IkkkbCA^{F/F}	High

8. Fig. 3e: why is the black line (control group) in 3e is different from the black line (control group) in 3g? Rather than the genotype effect (*Ikkkb*^{-/-} vs *Ikkkb*^{CA}), changes in controls appear to be making experimental groups show the difference (e.g. 38% vs 62% in controls at 6 h time point). Also, Fig. 3e and one data point in Fig. 3g have no error bars (or if the variance is minimal/close to none?).

Response: We thank the reviewer point out the difference of tau clearance capacity between the two control groups in Fig. 3. This is likely due to the lower seeding density (20,000 cells/well) of the control for *Ikkkb*^{CA} compared to the control for *Ikkkb*^{-/-} (25,000 cells/well). So even though both carry wildtype *Ikkkb* alleles, the tau degradation rates can differ (also see response #2 to Reviewer 1). We followed reviewer #1's suggestion to display all individual replicates for Fig. 3e,3g now. The error bars are SDs and can be visualized.

9. Supp.Fig.4g: why there is no AT8/MC1 in 4-5 month-old (if they are age-matched as other panels in the figure) PS19 mice (which should have quite a robust basal tau pathology at this age)?

Response: Based on the original paper and our extensive experiences with this model, MC1 immunoreactivity becomes prominent at age of 8–10 months ((Yoshiyama et al., 2007). Very few MC1 positive neurons can be detected before 6-month-old as shown by Holmes (Holmes et al., 2014).

10. Since this entire study hinges on NF-κB activation, it may be important to show panels of nuclear p-p65 staining in a few representative experiments (specifically for *in vivo* data sets) to conclusively establish that it is 'activation' of NFκB - not just overexpression of *Ikkkb* is driving tau spreading.

Response: Our transcriptome study demonstrated that *Ikkkb*^{CA} overexpression activated NF-κB signaling pathway (Fig 2e) and corresponding target genes (supplementary table 7) in primary microglia. To confirm that *Ikkkb*^{CA} overexpression activates NF-κB signaling *in vivo*, we analyzed the bulk RNA-seq data from *Ikkkb*^{CA} mice and predict the potential transcription factors using TRRUST database in Enrichr transcription panel (<https://maayanlab.cloud/Enrichr/>). NF-κB and RELA are predicted as the top transcription factor, providing direct evidence of elevated NF-κB activation in *Ikkkb*^{CA} mice. The new results were incorporated in new Supplementary Fig. 5.

11. In Fig. 4g-h, there should be more microglial MC1+ tau expected coupled with reduced neuronal MC1+ tau in the ipsilateral cortex of PS19/*Ikbkb*^{-/-} (per data from Fig. 3e-g). That means, the PS19 with *Ikbkb*^{-/-} background doesn't show the same phenotype in accumulating tau in microglia, as in vitro microglia in Fig. 3e-g? One way to confirm this is by doing a double IF for *Iba1*+MC1 or *Iba1*+AT8 for panels in Fig. 5c-d and/or Fig. 6a-c.

Response: In chronic conditions of human tauopathies and mouse models of tauopathies, there has been no report of appreciable tau inclusions in microglia, most likely due to highly efficient lysosomal degradation in these cells.

12. Fig. 4f-h shows cortex and k-m shows hippocampus. Can authors include both cortex and hippocampus in both panels for better comparison?

Responses: For *Ikbkb*^{-/-}; *P301S*⁺ and *Ikbkb*^{+/+}; *P301S*⁺ mice (Fig. 4f-h and rebuttal Fig. 1a), we inoculated tau fibrils at higher dose (5ug) in order to see tau spreading to cortex. We observed a trend of reduction of tau pathology with NF-κB inactivation although not reaching significance, in hippocampus of *Ikbkb*^{-/-}; *P301S*⁺ mice (rebuttal Fig. 1a). This might be due to relative strong tau signals in the hippocampus, which may mask the effect.

For *Ikbkb*^{CA}; *P301S*⁺ and *Ikbkb*^{WT}; *P301S*⁺ mice (Fig. 4k-m and rebuttal Fig 1b), we inoculated tau fibrils at lower dose (0.5ug) to avoid saturation effect. We did observe significant increase of tau seeding and spreading in hippocampus of *Ikbkb*^{CA}; *P301S*⁺ mice (Fig. 4k-m). However, due to the very low and highly variable spreading to cortical region (most data points of cortex MC1 occupied area are close to 0 as shown in rebuttal Fig. 1b), we did not detect significance comparing to *Ikbkb*^{WT}; *P301S*⁺ mice. A longer time course might be needed to see a significant effect.

Rebuttal Figure 1 Impacts of microglial NF-κB activation on tau seeding and spread in PS19 mice.

(a) Quantification of the percentage of MC1 occupied area in ipsilateral and contralateral hippocampus in *Ikbkb*^{+/+}; *P301S*⁺ (n=10) and *Ikbkb*^{-/-}; *P301S*⁺ (n=15) mice. 5 sections per mouse. values are mean ± SD. (b) Quantification of the percentage of MC1 occupied area in ipsilateral and contralateral cortex in *Ikbkb*^{WT}; *P301S*⁺ (n=9) and *Ikbkb*^{CA}; *P301S*⁺ (n=14) mice. 8 sections per mouse. values are mean ± SD

13. There seems to be some disconnect in terms of the consequences of what happens to tau seeds within microglia. In Fig. 3e-g, microglial NFκB activation (in case of *Ikbkb*^{CA} microglia), leads to reduced intra-microglial tau (i.e. increased tau processing - hopefully, clearance), which is not the case in microglia with inactivated NFκB (in case of *Ikbkb*^{-/-} MG). If so, then how the

results in Fig. 5 on the inactivation of microglial NF κ B showing a rescued phenotype – amidst build-up of tau within microglia per Fig. 3e-g?

Response: We would like to clarify that inactivation of microglial NF- κ B did not lead to tau buildup in microglia in vivo. The increased retention of tau in culture microglia by NF- κ B inactivation is likely due to reduced secretion by NF- κ B inactivation as shown in our new experimental data.

To further dissect the mechanisms, we investigated the effects of tau-mediated NF- κ B activation on chaperone-mediated autophagy (CMA) since CMA deficits was found to promote extracellular tau release (Caballero et al., 2021). We found that tau impairs CMA via activating NF- κ B and that inhibiting NF- κ B rescued CMA deficits in microglia (new Fig 6 g-k). our data support the model (new Fig. 8) that NF- κ B inhibition in microglia could exert neuroprotection by reducing pathogenic tau released from microglia. Further studies are needed to dissect how microglia communicate with neuron to modulate neuronal proteostasis.

14. The behavior tests were done at 8-9 months in Fig.5. While in Fig.6 (9-10 months), *Ikbkb*^{-/-} mice have increased tau inclusion. Are these the same mice used for the behavior tests in Fig. 5? Relevant to comment No. 9 above, microglial/neuronal markers should be used at the same time with MC1 or AT8 staining.

Response: Yes, the same mice were used for both behavior tests and imaging of tau inclusions. As mentioned above, no tau inclusions can be detected in microglia of human tauopathies or mouse tauopathies models, probably due to efficient lysosomal degradation in microglia in vivo.

15. Page 13, line 366. “NF- κ B activation could promote microglia to secrete more seeding-competent tau, and thus accelerate the spread of tau pathology.” To prove this statement, conditioned media from PS19/ *Ikbkb*^{CA} cells (and controls) should be investigated for 'seeding capable tau' in a seeding assay. As it is stated, there is no data to prove that PS19/ *Ikbkb*^{CA} microglia do secrete 'seedable tau'. Rather current results appear to achieve better clearance of tau in PS19/ *Ikbkb*^{CA} microglia.

Response: We thank the reviewer for helpful discussion and suggestions. To directly test the role of microglial NF- κ B in promoting secretion of seeding competent tau, we perform new experiments, and included in revised Fig. 3. We use a well-established reporter assay to determine the seeding capacity of microglial conditioned medium. We found that the conditional medium from microglia loaded with AD-tau show seeding activity in HEK biosensor cells even without transfection assistance of lipofectamine reagent.

16. Microglial NF κ B modulation can also affect synaptic pruning. Have the authors looked into investigating any synaptic markers that would support their behavioral readouts?

Response: We agree with the reviewer that microglial NF- κ B activity may influence the synapses, potentially via complement genes, which are target genes for NF- κ B. Systematic

analyses on how microglial NF- κ B affects the synapses of different subtypes of neurons are subject of our future studies.

Minor:

1. Page 3, line 68: “onset”, not “onsite”.

Response: Thanks for identifying this typo. We now corrected it in our revision.

2. Fig. 1g: the caption of the figure should be “stimulated”, not “stiumuted”.

Response: corrected.

References:

- Caballero, B., Bourdenx, M., Luengo, E., Diaz, A., Sohn, P.D., Chen, X., Wang, C., Juste, Y.R., Wegmann, S., Patel, B., *et al.* (2021). Acetylated tau inhibits chaperone-mediated autophagy and promotes tau pathology propagation in mice. *Nat Commun* 12, 2238.
- Holmes, B.B., Furman, J.L., Mahan, T.E., Yamasaki, T.R., Mirbaha, H., Eades, W.C., Belaygorod, L., Cairns, N.J., Holtzman, D.M., and Diamond, M.I. (2014). Proteopathic tau seeding predicts tauopathy in vivo. *Proc Natl Acad Sci U S A* 111, E4376-4385.
- Hopp, S.C., Lin, Y., Oakley, D., Roe, A.D., DeVos, S.L., Hanlon, D., and Hyman, B.T. (2018). The role of microglia in processing and spreading of bioactive tau seeds in Alzheimer's disease. *J Neuroinflammation* 15, 269.
- Kahlson, M.A., and Colodner, K.J. (2015). Glial Tau Pathology in Tauopathies: Functional Consequences. *J Exp Neurosci* 9, 43-50.
- Luo, W., Liu, W., Hu, X., Hanna, M., Caravaca, A., and Paul, S.M. (2015). Microglial internalization and degradation of pathological tau is enhanced by an anti-tau monoclonal antibody. *Sci Rep* 5, 11161.
- Pérez, M., Avila, J., and Hernández, F. (2019). Propagation of Tau via Extracellular Vesicles. *Front Neurosci* 13, 698.
- Yoshiyama, Y., Higuchi, M., Zhang, B., Huang, S.M., Iwata, N., Saito, T.C., Maeda, J., Suhara, T., Trojanowski, J.Q., and Lee, V.M. (2007). Synapse loss and microglial activation precede tangles in a P301S tauopathy mouse model. *Neuron* 53, 337-351.

Reviewers' Comments:

Reviewer #2:

Remarks to the Author:

The authors have made numerous changes in the revision and have addressed prior concerns.

Reviewer #3:

Remarks to the Author:

I really like to acknowledge that Dr. Gan and her team have done an excellent job in revising the manuscript as suggested by all three reviewers and answering all prior concerns with new experiments and coming up with a new hypothetical model for their study. This reviewer has no further concerns or questions.

Reviewer #4:

Remarks to the Author:

In the revised manuscript, Wang et al investigated the role of NF- κ B in driving tau seeding and associated tau associated neurotoxicity. The authors have conducted comprehensive study that demonstrated the role of microglia-associated neuroinflammation in driving Tauopathy. Authors have been very responsive and addressed previous comments point-by-point, and these answers are proper and adequate. By utilizing in vitro and in vivo mouse models including microglial specific downregulation of *Ikbkb*^{-/-} and tau model with constitutively active *Ikbkb*^{CA}, authors clearly and showed that microglial NF- κ B has a role in processing and propagating Tau spread by regulating proteolysis and exocytosis. The study has large sets of experiments data, and analyses are rigor. The conclusion is strongly supported by their results and highly important. The manuscript is written clearly and easy to follow, despite the dataset is very condensed.

Some minor corrections,

1) Fig. 5j, please correct the y axis labelling typo.

2) Fig. 7, please include the no. of microglia specific single nuclei per genotype.

3) In the discussion section authors mention about "inactivating microglial NF- κ B protected against tau-mediated cognitive deficits despite elevated tau inclusions reveals that tau pathology and toxicity are multifaceted and cell type-specific". Since cross talk between microglia and neurons affect the cognitive function and given that single nuclei RNA seq data set provided in Fig. 7 have large no. of excitatory and inhibitory neurons, it will be more informative to provide brief discussion about what kind of neuronal genes were altered, and how they might be associated with tau-mediated hippocampus memory deficits.

Reviewer #5:

Remarks to the Author:

Microglia undergo extensive transcriptomic reprogramming in various neurodegenerative diseases, including Alzheimer's disease (AD). The major questions that remain unaddressed include how the transcriptomic reprogramming in microglia is controlled, and whether the altered reprogramming in microglial transcriptome contributes to their functional deterioration in AD. In this study, Wang et al. reported a novel role of NF- κ B-dependent transcriptomic reprogramming in microglia, which is shown to regulate tau pathogenesis in a mouse model of tauopathy. The authors first showed that tau induces the expression of microglial genes that are known targets for the transcription factor NF- κ B. Through genetic manipulation of NF- κ B expression and its activity, they demonstrated that overactivation of NF- κ B promotes tau seeding and spreading, while genetic ablation of microglial NF- κ B attenuates tau seeding and spreading. In line with this, genetic ablation of NF- κ B rescues the cognitive impairment in the mouse model of tauopathy. This study provides an important insight on how tau triggers transcriptomic reprogramming in microglia,

which in turn regulates the pathogenesis of tau.

To further substantiate the conclusions of this study, the authors should address the following minor comments.

1. In Figures 1A and 1E, the authors should indicate the similarity between tau-induced transcriptomic changes in microglia in vitro and in vivo.
2. In Figures 1I and 1J, it appears that the IPA analysis was performed using human protein reference (for example the protein HLA-A refers to a subunit of human MHC complexes). The authors should use mouse protein reference if all these experiments were conducted in the mouse system.
3. In Figure 2B, measurement of the mRNA level of NF- κ B alone could not reflect whether the transcription factor is constitutively active. The authors need to at least address whether there is an increased nuclear accumulation of NF- κ B.
4. In Figures 2C and 2H, the authors should clearly indicate how the list of DEGs are generated (i.e. *IkkkbCA* vs. *IkkkbWT*, instead of only depicting *IkkkbCA*).
5. While the data from Figures 3A to 3G seem to support that NF- κ B promotes the degradation of tau fibrils, the authors need to demonstrate that the genetic ablation or overactivation of NF- κ B will not influence the uptake of tau fibrils at 0 h.
6. In Figures 3E and 3G, the rates of tau fibril processing for *Ikkkb*^{-/-} (Figure 3E) and *IkkkbWT* (Figure 3G) mice are both at ~40%. Moreover, microglial culture from *Ikkkb*^{+/+} and *IkkkbWT* are morphologically distinctive (Figures 3D and 3F). The authors need to characterize the basal phenotypes to support proper comparisons, e.g. a bulk RNA-seq analysis of microglia derived from *Ikkkb*^{+/+} and *IkkkbWT*.
7. In Figure 7, the authors should list out top signature genes from each microglial subcluster in Figure 7C.
8. The rationale for Figure 7E is unclear, is the pseudotime indicating the microglial developmental lineage induced by tau? If so, why would the microglia from *IkkkbCA*; P301S be included?

Point-to-point response to the reviewers:

Reviewer #2

The authors have made numerous changes in the revision and have addressed prior concerns.

We thank the reviewer for their positive comments.

Reviewer #3

I really like to acknowledge that Dr. Gan and her team have done an excellent job in revising the manuscript as suggested by all three reviewers and answering all prior concerns with new experiments and coming up with a new hypothetical model for their study. This reviewer has no further concerns or questions.

We thank the reviewer for their positive comments.

Reviewer #4

In the revised manuscript, Wang et al investigated the role of NF- κ B in driving tau seeding and associated tau associated neurotoxicity. The authors have conducted comprehensive study that demonstrated the role of microglia-associated neuroinflammation in driving Tauopathy. Authors have been very responsive and addressed previous comments point-by-point, and these answers are proper and adequate. By utilizing in vitro and in vivo mouse models including microglial specific downregulation of *Ikbkb*^{-/-} and tau model with constitutively active *Ikbkb*^{CA}, authors clearly and showed that microglial NF- κ B has a role in processing and propagating Tau spread by regulating proteolysis and exocytosis. The study has large sets of experiments data, and analyses are rigor. The conclusion is strongly supported by their results and highly important. The manuscript is written clearly and easy to follow, despite the dataset is very condensed.

We thank the reviewer for the positive comments.

Some minor corrections,

1) Fig. 5j, please correct the y axis labelling typo.

Corrected.

2) Fig. 7, please include the no. of microglia specific single nuclei per genotype.

Response: We now include the number of microglia specific single nuclei of each genotype in the Result part as following:

*'We then examined the trajectory of the subclusters of a total 4305 microglia (1068 from *Ikbkb*^{+/+}, 1275 from *Ikbkb*^{+/+};*P301S*⁺, 720 from *Ikbkb*^{-/-};*P301S*⁺ and 1242 from *Ikbkb*^{CA};*P301S*⁺) to investigate how NF- κ B affects microglial states in *P301S* mice using Monocle'*

3) In the discussion section authors mention about "inactivating microglial NF- κ B protected against tau-mediated cognitive deficits despite elevated tau inclusions reveals that tau pathology and toxicity are multifaceted and cell type-specific". Since cross talk between microglia and neurons affect the cognitive function and given that single nuclei RNA seq data set provided in Fig. 7 have large no. of excitatory and inhibitory neurons, it will be more informative to provide brief discussion about what kind of neuronal genes were altered, and how they might be associated with tau-mediated hippocampus memory deficits.

Response: Thanks for this excellent suggestion. We have included the IPA canonical pathways of excitatory and inhibitory neurons altered by *Ikbkb* deletion in *P301S* mice as supplementary figure 10. We also discussed in Discussion section as following:

'Single nuclei RNA-seq analyses of transcriptomes of excitatory and inhibitory neurons revealed that inactivation of microglial NF-κB caused marked changes in these neurons (Supplementary Fig. 10). Microglial NF-κB inactivation led to elevated synaptogenesis pathways in excitatory neurons, and endocannabinoid synapse and P2Y purigenic receptor signaling pathways in inhibitory neurons, consistent with the protective effects of inactivating NF-κB.'

Reviewer #5

Microglia undergo extensive transcriptomic reprogramming in various neurodegenerative diseases, including Alzheimer's disease (AD). The major questions that remain unaddressed include how the transcriptomic reprogramming in microglia is controlled, and whether the altered reprogramming in microglial transcriptome contributes to their functional deterioration in AD. In this study, Wang et al. reported a novel role of NF-κB-dependent transcriptomic reprogramming in microglia, which is shown to regulate tau pathogenesis in a mouse model of tauopathy. The authors first showed that tau induces the expression of microglial genes that are known targets for the transcription factor NF-κB. Through genetic manipulation of NF-κB expression and its activity, they demonstrated that overactivation of NF-κB promotes tau seeding and spreading, while genetic ablation of microglial NF-κB attenuates tau seeding and spreading. In line with this, genetic ablation of NF-κB rescues the cognitive impairment in the mouse model of tauopathy. This study provides an important insight on how tau triggers transcriptomic reprogramming in microglia, which in turn regulates the pathogenesis of tau.

We thank the reviewer for the positive comments.

To further substantiate the conclusions of this study, the authors should address the following minor comments.

1. In Figures 1A and 1E, the authors should indicate the similarity between tau-induced transcriptomic changes in microglia in vitro and in vivo.

Response: We compared DEGs in Figures 1A and 1E and found that nearly 30% of the DEGs in PS19 microglia were also changed in tau stimulated microglia, including DAM genes and NF-κB target genes. The full list of DEGs is included in new Supplementary Table 5. Selected genes are highlighted with blue text in Figure 1A and 1E. We also incorporated a description in the Results section as following:

"59 of 187 upregulated DEGs in PS19 microglia, including DAM genes (e.g. Clec7a, Cst7, Lpl) and NF-κB target genes (e.g. Tnfa, Il1rn, Tlr2), were also upregulated in tau fibrils stimulated microglia, indicating a similarity between tau-induced transcriptomics changes in microglia in vitro and in vivo.(Fig. 1a, 1e and supplementary table 5)."

2. In Figures 1I and 1J, it appears that the IPA analysis was performed using human protein reference (for example the protein HLA-A refers to a subunit of human MHC complexes). The authors should use mouse protein reference if all these experiments were conducted in the mouse system.

Response: In Figures 1I and 1J, each symbol represents a differentially expressed gene but not the protein. We thank the reviewer point out the incorrect output of IPA analysis. We confirmed that mouse Ensembl ID (ENSMUSGxxxxxxx) were used as identifier in IPA data uploading and mouse specie was selected as filter for core analysis. In revised Figure 1I and 1J, we changed the symbol to the correct mouse gene format (italicized, with only the first letter in upper-case) and make sure the gene names are mouse orthologs (e.g. H2-D1 instead of HLA-A). The same revisions are also made to Figures 1A and 1E.

3. In Figure 2B, measurement of the mRNA level of NF-κB alone could not reflect whether the

transcription factor is constitutively active. The authors need to at least address whether there is an increased nuclear accumulation of NF- κ B.

Response: In addition to the increase of the mRNA level of NF- κ B, we have provided multiple evidences to support the activation of NF- κ B in our manuscript. Our transcriptome study clearly demonstrated that, in vitro, *Ikkkb* CA overexpression activated NF- κ B signaling pathway (Fig 2e, Supplementary Fig 2b) and corresponding target genes (supplementary table 7, table 9) in primary microglia; in vivo, NF- κ B and RELA are predicted as the top transcription factor, providing direct evidence of elevated NF- κ B activation in *Ikkkb* CA mice.

4. In Figures 2C and 2H, the authors should clearly indicate how the list of DEGs are generated (i.e. *Ikkkb*CA vs. *Ikkkb*WT, instead of only depicting *Ikkkb*CA).

Response: We now indicate the corresponding control for each list of DEGs in Figures 2C and 2H and Figure 6E.

5. While the data from Figures 3A to 3G seem to support that NF- κ B promotes the degradation of tau fibrils, the authors need to demonstrate that the genetic ablation or overactivation of NF- κ B will not influence the uptake of tau fibrils at 0 h.

Response: In Figures 3A to 3G, the cells were incubated continuously with tau fibrils for 24hrs allow us to study tau processing by microglia, which we believe mimic the chronic exposure to tau in disease and is our major focus in this study.

6. In Figures 3E and 3G, the rates of tau fibril processing for *Ikkkb*^{-/-} (Figure 3E) and *Ikkkb*WT (Figure 3G) mice are both at ~40%. Moreover, microglial culture from *Ikkkb*^{+/+} and *Ikkkb*WT are morphologically distinctive (Figures 3D and 3F). The authors need to characterize the basal phenotypes to support proper comparisons, e.g. a bulk RNA-seq analysis of microglia derived from *Ikkkb*^{+/+} and *Ikkkb*WT.

Response: *Ikkkb*^{+/+} and *Ikkkb*^{WT} both carry wildtype *Ikkkb* alleles, and no morphological difference was observed.

Direct comparisons between Figure 3E and Figure 3G cannot be made because the seeding densities were different for the phagocytosis assays. We plated fewer cells in the study of *Ikkkb*^{WT} vs. *Ikkkb*^{CA} (20,000 cells/well, Figure 3G) and more cells in the study of *Ikkkb*^{-/-} vs *Ikkkb*^{+/+} (25,000 cells/well, Figure 3E). So the overall tau processing rates are faster in Figure 3E. For assays in Figure 3E and 3G, we performed three to four independent experiments with 4-8 biological replicates and obtained consistent results.

7. In Figure 7, the authors should list out top signature genes from each microglial subcluster in Figure 7C.

Response: We have included the top signature genes from each microglia subcluster in Supplementary table_15

8. The rationale for Figure 7E is unclear, is the pseudotime indicating the microglial developmental lineage induced by tau? If so, why would the microglia from *Ikkkb*CA; P301S be included?

Response: We included the microglia from *Ikkkb*^{CA}; P301S to directly compare with *Ikkkb*^{-/-}, P301S to gain insight into the interplay between microglial pathways induced by tau vs. those directly activated by NF- κ B. Our analysis showed that while deletion of *Ikkkb* prevented the progression of microglia induced by tau, microglia from *Ikkkb*^{CA};P301S+ mice extends beyond the tau-associated disease states (cyan, Fig. 7f), suggesting a positive feedback mechanism between tau and NF- κ B activation in microglia. We have clarified further in the discussion as following:

“Constitutive NF- κ B activation in PS19 mice further extended microglial states beyond the tau-associated disease states, supporting a feed forward mechanism between tau toxicity and microglial NF- κ B activation (Fig. 8)”.

Reviewers' Comments:

Reviewer #5:

Remarks to the Author:

The authors have made substantial revisions to address my previous concerns.

REVIEWERS' COMMENTS

Reviewer #5 (Remarks to the Author):

The authors have made substantial revisions to address my previous concerns.

Response: We appreciate the reviewers' comments.

Editor's COMMENTS for human research participants (on Summary Reports)

1. Please ensure only one department or institution is listed in a single affiliation.

Response: Affiliations are edited, except for "Helen and Robert Appel Alzheimer's Disease Institute, Brain and Mind Research Institute, Weill Cornell Medicine, New York, NY, USA", we need to keep Brain and Mind Research Institute, which is our department, part of our affiliation.

2. We observed human brain tissues are involved in the study. Please check the box "involved in the study" and provide the necessary information in the appropriate module. Also, please provide statement on ethical approval along with full name of ethics committee for use of Frontal cortical tissue from AD individual, here in the reporting summary as well as in the manuscript.

Response: The human brain samples used in this study are not considered identified "human subjects", thus the institutional review boards of Weill Cornell Medicine and UCSF have determined that the clinicopathologic studies on de-identified postmortem tissue samples are exempt from human subject research according to Exemption 45 CFR 46.104(d)(2). Thus, we did not check the box in the reporting summary. But we have provided the following statement in our manuscript:

Method section->Isolation of the sarkosyl-insoluble fraction from human brain (AD-tau)

"Frontal cortical tissue from AD individual was obtained from the New York Brain bank of Columbia University and are not considered identified "human subjects" and not subjected to IRB oversight. All brains were donated after consent from the next of- kin or an individual with legal authority to grant such permission. The institutional review board of Weill Cornell Medicine has determined that clinicopathologic studies on de-identified postmortem tissue samples are exempt from human subject research according to Exemption 45 CFR 46.104(d)(2)."

Method section-> Preparation of PSP brain extract

"Human brain tissues from PSP cases, which were diagnosed based on accepted neuropathology criteria, were obtained from UCSF Memory and Aging Center. All brains were donated after consent from the next of- kin or an individual with legal authority to grant such permission. The use of postmortem brain tissues for research was approved by the University of California, San Francisco's Institutional Review Board with informed consent from patients or their families. The institutional review board has determined that clinicopathologic studies on de-identified

postmortem tissue samples are exempt from human subject research according to Exemption 45 CFR 46.104(d)(2).”